# Ground-based remote sensing of $O_3$ by high and medium resolution FTIR spectrometers over the Mexico City basin

Eddy F. Plaza-Medina[1], Wolfgang Stremme[1], Alejandro Bezanilla[1], Michel Grutter[1], Matthias Schneider[2], Frank Hase[2], and Thomas Blumenstock[2]

[1]Centro de Ciencias de la Atmósfera, Universidad Nacional Autónoma de México, 04510 Ciudad de México, México
[2]Institute of Meteorology and Climate Research (IMK-ASF), Karlsruhe Institute of Technology, Karlsruhe, Germany

*Correspondence to:* E. F. Plaza-Medina and M. Schneider
(eddymedi@gmail.com and matthias.schneider@kit.edu)

**Abstract.** We present atmospheric ozone ($O_3$) profiles measured over central Mexico between November 2012 and February 2014 by two different ground-based FTIR (Fourier transform infrared) solar absorption experiments. The first instrument offers very high resolution spectra and contributes to NDACC (Network for the Detection of Atmospheric Composition Change). It is located at a mountain observatory about 1700 m above the Mexico City basin. The second instrument has a medium spectral resolution and is located inside of Mexico City at a horizontal distance of about 60 km to the mountain observatory. It is documented that the retrieval with the high and medium resolution experiments give $O_3$ variations for four and three independent atmospheric altitude ranges, respectively, whereby the theoretically estimated errors of these profile data are mostly within 10%. The good quality of the data is empirically demonstrated: above the tropopause by intercomparing the two FTIR $O_3$ data and for the boundary layer by comparing the Mexico City FTIR $O_3$ data with in-situ $O_3$ surface data. Furthermore, we develop a combined boundary layer $O_3$ remote sensing product that uses the retrieval results of both FTIR experiments and use theoretical and empirical evaluations for documenting the improvements that can be achieved by such combination.

## 1 Introduction

$O_3$ is an important atmospheric trace gas. In the stratosphere $O_3$ absorbs ultraviolet (UV) radiation thereby protecting all living organisms. During the second half of the last century stratospheric ozone has experienced significant depletion. With the implementation of the Montreal Protocol in 1987 and its amendments the anthropogenic emissions of the mainly responsible chlorine and fluoride compounds were strongly restricted and stratospheric $O_3$ depletion has progressively diminished (Scientific Assessment of Ozone Depletion, World Meteorological Organization, Pawson et al., 2014), and it is expected to become negligible by the end of this century (Hegglin and Shepherd, 2009). In the troposphere $O_3$ is an important pollutant and greenhouse gas (Jacob and Winner, 2009; Stock et al., 2014) and has increased since pre-industrial times (IPCC-2013, 2013). $O_3$ is hazardous to human health and a severe problem in many cities around the globe as it is produced by complex photochemistry in presence of anthropogenic pollutants ($NO_x$, CO, and VOC) and UV radiation (e.g. Brasseur et al., 1999).

For understanding the stratospheric or tropospheric $O_3$ evolution the whole atmosphere has to be considered. For instance, Ramaswamy et al. (2006) showed that there is a correlation between the stratospheric cooling, stratospheric $O_3$ depletion and the increase of greenhouse gases (mainly anthropogenic $CO_2$). Increases in tropospheric greenhouse gas concentrations warm the lower and middle troposphere and cool the stratosphere. The changed vertical temperature structure affects atmospheric dynamics and stratospheric $O_3$ distribution. The complexity of these interactions make it difficult to predict how, when, and to what extent stratospheric $O_3$ recovery will take place (Weatherhead and Andersen, 2006). Because tropospheric $O_3$ is a greenhouse gas, increasing tropospheric $O_3$ levels will have an effect on the evolution of stratospheric $O_3$ concentrations. Similarly, tropospheric $O_3$ levels are influenced by the stratosphere and stratospheric $O_3$ concentrations in different ways. A direct connection exists during stratosphere-troposphere exchange events. Furthermore, stratospheric $O_3$ or aerosol loading affect the UV radiation penetrating to the troposphere and consequently the photochemistry of tropospheric $O_3$ and other species (Zhang et al., 2014).

A comprehensive investigation of these complex climate-chemistry interactions and the particular role of $O_3$ is only possible by combining atmospheric models with observations (Dameris and Jöckel, 2013; Hassler et al., 2013). In this context ground-based high resolution solar absorption FTIR spectrometer measurements have been proven to be useful, providing information of the vertical distribution of $O_3$ and other trace gases (e.g. Rinsland et al., 2003; Kohlhepp et al., 2012; Vigouroux et al., 2015; Barthlott et al., 2017). High resolution and high quality FTIR measurements are organised in the framework of NDACC and are made at about 20 globally distributed sites (www.acom.ucar.edu/irwg/). Due to the high quality and long-term characteristics, the NDACC FTIR data are very interesting for trend studies (e.g. Kohlhepp et al., 2012; García et al., 2012; Vigouroux et al., 2015). However, the NDACC-like ground-based FTIR measurements are mainly performed at remote sites (i.e. far away from polluted areas) and very scarce in tropical latitudes. The only NDACC-like FTIR sites reporting atmospheric data within or close to the 20°N-20°S latitude belt are Mauna Loa (19.5°N, 155.6°W), Addis Ababa (9.0°N, 38.8°E, Takele Kenea et al., 2013), Paramaribo (5.8°N, 55.2°W, Petersen et al., 2008) and La Réunion (21.1°S, 55.4°E, Senten et al., 2008). Medium resolution FTIR spectrometers have also been used to measure atmospheric $O_3$ amounts (e.g. over the megacity of Paris, Viatte et al., 2011). However, a comprehensive quality documentation of the $O_3$ data retrieved from medium resolution FTIR spectra is still missing.

Here we present atmospheric $O_3$ profiles obtained from solar absorption spectra, measured by two different FTIR spectrometers in central Mexico (at about 19°N, 99°W). The first instrument is a high resolution FTIR spectrometer located at a high altitude station (Altzomoni), but very close to Mexico City. In the meanwhile these FTIR activities form part of NDACC and together with the Paramaribo measurements represent the only NDACC FTIR contribution from Latin America. The second instrument is a medium resolution FTIR spectrometer located inside of Mexico City, a megacity whose emissions have been investigated in great detail in the context of MILAGRO (Megacities Initiative: Local And Global Research Observations, Molina et al., 2010, and references therein). We would like to point out that ground-based remote sensing measurements of the vertical $O_3$ distribution between the boundary layer and the upper stratosphere are challenged by the fact that the $O_3$ concentrations strongly vary with altitude. This is shown in the left panel of Fig. 1, which shows a typical tropical $O_3$ profile (blue line for volume mixing ratios and black for abundances relative to total abundances).

The focus of this paper is to demonstrate the reliability of the FTIR $O_3$ profiles between the boundary layer and the upper stratosphere and in the following sections we will have a closer look on the altitude regions marked by yellow stars in the left panel of Fig.1. Section 2 presents the high and medium resolution FTIR remote sensing experiments, the remote sensing retrievals and discusses the main characteristics of the retrieval products. In Sect. 3 we analyse the $O_3$ concentrations above the boundary layer and show that the total column amounts and lower and middle stratospheric $O_3$ amounts retrieved from the medium resolution FTIR measurements are in good agreement to the amounts retrieved from the high resolution FTIR measurements. In Sect. 4 we investigate the possibility of observing $O_3$ concentrations in the boundary layer by the medium resolution FTIR instrument that is located in Mexico City. We empirically prove this possibility using boundary layer in-situ $O_3$ data as a reference and show that by combining the two remote sensing experiments we can generate a boundary layer $O_3$ remote sensing product with improved precision. Section 5 gives a summary and outlook.

## 2    Ground-based FTIR remote sensing

The right panel of Fig. 1 indicates the location of the two ground-based FTIR remote sensing and the three in-situ surface instruments that are used in the context of Sect. 4. The first FTIR instrument is located in Altzomoni within the Izta-Popo National Park at nearly 4000 m a.s.l. between the Popocatépetl and the Iztaccílhuatl volcanoes at 19.12°N, 98.65°W, about 1700 m above the Mexico City basin. The second instrument is located at the UNAM (Universidad Nacional Autónoma de México) in Mexico City, at the rooftop of the CCA (Centro de Ciencias de la Atmósfera, 19.33°N, 99.18°W, 2260 m a.s.l). The UNAM stations is at a horizontal distance of about 60 km to Altzomoni in northwest direction and both stations are operated by the Spectroscopy and Remote Sensing Group of CCA/UNAM.

### 2.1    Experimental setup

A ground-based FTIR remote sensing experiment consists of a solar tracker and a Michelson-type interferometer. The solar tracker captures the direct solar light beam and couples it into the interferometer, which splits the solar light into two beams. The first beam traverses a fixed distance and the second beam a variable distance. Finally the intensity of the recombined beam is analysed by a detector. In the middle infrared, where the here used two instruments are measuring, photo-conductive as well as photo-voltaic detectors are applied. The intensity of the recombined beam depend on their optical path difference and these intensity signals are called the interferogram. The spectrum is then calculated by a Fourier transformation of the interferogram.

The maximum optical path difference, the apodisation, and misalignments of the interferometer (which can be minimised but not completely avoided) determine the instrumental line shape (ILS, the response of the instrument to a monochromatic input radiation). Understanding the ILS is important for correctly interpreting the measured spectra. The effect of misalignments on the ILS can be determined by low pressure gas cell measurements. Since the pressure and the absorption characteristics of the gas in the cell is known we can retrieve the ILS from such measurements (Hase, 2012).

## 2.2 Retrieval method

The measured solar absorption spectra (expressed as spectral bin vector $\boldsymbol{y}$) and the atmospheric state (expressed by atmospheric state vector $\boldsymbol{x}$) are connected via a radiative transport model (forward model $\boldsymbol{F}$):

$$\boldsymbol{y} = \boldsymbol{F}(\boldsymbol{x}, \boldsymbol{p}). \tag{1}$$

Here the vector $\boldsymbol{p}$ represents auxiliary atmospheric parameters (like temperature) or instrumental characteristics (like the instrumental line shape). Generally, there are many different atmospheric states $\boldsymbol{x}$ that can equally well explain the measured spectrum $\boldsymbol{y}$ within its measurement noise level. This means that we face an ill-posed problem and we need to constrain the solution state. The constraint solution is at the minimum of the cost function:

$$[\boldsymbol{y} - \boldsymbol{F}(\boldsymbol{x}, \boldsymbol{p})]^T \mathbf{S}_\epsilon^{-1} [\boldsymbol{y} - \boldsymbol{F}(\boldsymbol{x}, \boldsymbol{p})] + [\boldsymbol{x} - \boldsymbol{x}_a]^T \mathbf{S}_\mathbf{a}^{-1} [\boldsymbol{x} - \boldsymbol{x}_a]. \tag{2}$$

Here the first term is a measure of the difference between the measured spectrum ($\boldsymbol{y}$ with $\mathbf{S}_\epsilon$ capturing the measurement noise covariance) and the spectrum simulated for a given atmospheric state ($\boldsymbol{x}$). The second term is the regularisation term. It constrains the atmospheric solution state ($\boldsymbol{x}$) towards the a priori state $\boldsymbol{x}_a$, whereby the kind and the strength of the constraint are defined by the covariance matrix $\mathbf{S}_\mathbf{a}$. For more details about solving ill-posed problems in atmospheric remote sensing please refer to Rodgers (2000).

Since we face a non-linear problem the solution is reached iteratively, whereby for the $(i+1)$ iteration's step it is:

$$\boldsymbol{x}_{i+1} = \boldsymbol{x}_a + \mathbf{S}_\mathbf{a} \mathbf{K}_\mathbf{i}^T (\mathbf{K}_\mathbf{i} \mathbf{S}_\mathbf{a} \mathbf{K}_\mathbf{i}^T + \mathbf{S}_\epsilon)^{-1} [\boldsymbol{y} - \boldsymbol{F}(\boldsymbol{x}_i) + \mathbf{K}_\mathbf{i}(\boldsymbol{x}_i - \boldsymbol{x}_a)]. \tag{3}$$

Here $\mathbf{K}$ is the Jacobian matrix which samples the derivatives $\partial y / \partial x$ (changes in the spectral radiances $\boldsymbol{y}$ for changes in the vertical distribution of the atmospheric constituents $\boldsymbol{x}$).

An important addendum of the retrieved solution vector is the averaging kernel matrix $\mathbf{A}$. It samples the derivatives $\partial x / \partial x_{\mathrm{act}}$ (changes in the retrieved concentration $x$ for changes in the actual atmospheric concentration $x_{\mathrm{act}}$) describing the smoothing of the actual atmospheric state by the remote sensing measurement process:

$$(\boldsymbol{x} - \boldsymbol{x}_a) = \mathbf{A}(\boldsymbol{x}_{\mathrm{act}} - \boldsymbol{x}_a). \tag{4}$$

The matrix $\mathbf{A}$ can be calculated as:

$$\mathbf{A} = \mathbf{G}\mathbf{K} = (\mathbf{K}\mathbf{S}_\epsilon^{-1}\mathbf{K}^T + \mathbf{S}_\mathbf{a}^{-1})^{-1}\mathbf{K}^T\mathbf{S}_\epsilon^{-1}\mathbf{K}. \tag{5}$$

Here $\mathbf{G}$ is the gain matrix, which samples $\partial x / \partial y$ (changes in the retrieved state $\boldsymbol{x}$ for changes in the spectral radiances $\boldsymbol{y}$). The kernels are rather important since they document what is actually measured by the remote sensing system. Without this information, the remote sensing data cannot be used in a sensible manner. In addition, the trace of $\mathbf{A}$ quantifies the amount of information obtained by the measurement. It can be interpreted as the degrees of freedom of signal (DOFS) of the measurement.

In this study we work with the PROFFIT retrieval code (Hase et al., 2004), which has been used for many years by the ground-based FTIR community for evaluating high resolution solar absorption spectra. PROFFIT offers three kind of constrains

for solving the inversion problem: scaling of a priori profiles, Optimal Estimation (using an $\mathbf{S_a}$ constructed from a model or measurement climatology, Rodgers, 2000) and Tikhonov-Phillips (using an ad-hoc created $\mathbf{S_a}^{-1}$ for constraining towards absolute profile values or the profile shape, Tikhonov, 1963; Rodgers, 2000).

We use a model atmosphere with 41 and 44 discretised grid levels from the surface up to 120 km for the Altzomoni and UNAM retrievals, respectively, whereby the grid levels 1 to 3 of UNAM are situated below Altzomoni and the grid levels 4 to 44 of UNAM are the same as the grid levels 1 to 41 of Altzomoni. The $O_3$ inversions are regularised by a Tikhonov-Phillips constraint towards the vertical profile slope and towards the absolute value for the uppermost atmospheric model altitude (i.e. the 120 km grid level). The strength of the constraint is determined by starting with a weak constraint and then increasing it until a significant increase in the residual of the spectral fit is observed (L-curve criterion). The inversion is made on the logarithmic scale of $O_3$ volume mixing ratios (Hase et al., 2004; Schneider et al., 2006). As a priori volume mixing ratio profiles ($x_a$) for $O_3$ and all interfering species we use climatologies from the Whole Atmosphere Community Climate Model (WACCM, version 6). The $O_3$ a priori profile is depicted in the left panel of Fig. 1. The temperature and pressure profiles are from the National Centers for Environmental Prediction (NCEP) analysis. For altitudes above 50 km we use monthly mean CIRA temperature climatologies (Rees et al., 1990). Spectroscopic line parameters for $O_3$ and interfering species are taken from the High Resolution Transmission (HITRAN) 2008 database (Rothman et al., 2009), except for water, for which the HITRAN 2009 update is used.

## 2.3 Error analyses

The errors are estimated in form of a error covariance matrix $\mathbf{S_e}$:

$$\mathbf{S_e} = \mathbf{G}\mathbf{K_p}\mathbf{S_p}\mathbf{K_p}^T\mathbf{G}^T. \tag{6}$$

Here $\mathbf{G}$ is the gain matrix, $\mathbf{K_p}$ are the Jacobians with respect to the parameters $p$ (changes of spectral bins due to changes in the parameters $p$) and $\mathbf{S_p}$ is the uncertainty covariance for the parameters $p$. We assume uncertainties for the parameters as listed in Table 1, whereby assuming that the uncertainties between the different parameters are uncorrelated.

We consider a white noise in the measured spectra of 0.3% and 0.5% (ratio between noise and highest intensity in the measured spectral region), which is what we typically observe in the Altzomoni and UNAM spectra. As baseline channeling amplitude we assume 0.2%, which is also what we occasionally observe in NDACC FTIR spectra measurements. We assume a relatively high baseline offset of 1% (higher than for other error assessment studies, e.g. Schneider and Hase, 2008), because in routine operations the spectra are often taken of sky that are partially covered by clouds. Generally, clouds cause very noisy measurements and can be easily identified. However, occasionally there might be clouds in the line of sight for a few seconds during a measurement (a measurement takes several minutes). Interferograms recorded during such intensity fluctuations then lead to spectra with an increased baseline offset. For the ILS we assume an uncertainty in form of a linear decay of the modulation from the zero path difference to maximal optical path difference by 5% and for the phase error 0.1 rad for all positions of the interferometer mirror. The atmospheric temperature uncertainty from NCEP is assumed to be within 1.5 K (from the surface up to 45 km, whereby we separately consider tropospheric and stratospheric uncertainties) and 6 K for higher

altitudes. As uncertainty of the solar tracker we use $0.2°$, which is a rather conservative estimate, since it is close to the radius of the solar disc (an error close to the radius of $0.25°$ would become clearly visible by high noise levels in the measured spectra). For estimating the effect of solar lines we work with the uncertainties that have been used in many other assessment studies (e.g. García et al., 2012). The spectroscopic parameters have typical uncertainties of 2 and 5% for line intensity and pressure

5 broadening, respectively (typical values as given in the HITRAN line parameter lists). Finally, we assume a 100% uncertainty in the atmospheric water vapour content. The large variability of tropospheric water vapour might affect the $O_3$ retrievals, even though the selected spectral windows contain only weak spectroscopic water vapour signatures.

## 2.4 Observations at Altzomoni

Infrared solar absorption spectra have been recorded at Altzomoni since May 2012 with a Bruker IFS 120/5HR spectrometer

10 (an IFS 120HR which has been upgraded with the electronics of an IFS 125HR). The sunlight is followed and guided to the spectrometer with a solar tracker, which is equipped with two plane ellipse-shape mirrors and two motors of rotation stage, one to access different elevation angles and the other to reach azimuthal directions. The pointing of the solar tracker is monitored and controlled with a standard CMOS USB-camera and the Camtracker software (Gisi et al., 2011). The solar tracker is protected with a motorized dome and the FTIR spectrometer and solar tracker are operated remotely from the UNAM

15 campus in Mexico City. The spectrometer allows measurements with very high resolution ($0.0035\,\mathrm{cm}^{-1}$, i.e. maximum optical path difference of 257 cm) and is equipped with two beam-splitters: a potassium bromide (KBr, which is normally used) and a calcium fluoride ($CaF_2$, which is only occasionally used). Routine measurements are performed with three detectors: mercury-cadmium-telluride (MCT), indium-antimonide (InSb) and indium-gallium-arsenide (InGaAs). The first two are nitrogen-cooled and cover the spectral range of $700\text{-}4200\,\mathrm{cm}^{-1}$ (mid-infrared region) and the third detector works at room temperature in the

20 spectral range $4000\text{-}12800\,\mathrm{cm}^{-1}$ (near infrared region). In order to increase the signal to noise ratio in the mid-infrared spectral region, six NDACC-type filters are used to cover different spectral sub-regions. The Altzomoni FTIR experiment forms part of NDACC (www.ndacc.org) and also contributes to the MUSICA activities (Barthlott et al., 2017).

In order to investigate a potential misalignment of the interferometer and interpret the observed line shapes of the solar absorption spectra in a correct manner (Hase, 2012), we obtained the ILS from HBr cell measurements. This cell has a length

25 of 2 cm and is filled with HBr at a nominal pressure of 200 Pa (total pressure is between 200 to 250 Pa). The modulation and phase error of the ILS were calculated from the cell measurements using the LINEFIT code (Hase et al., 1999). The estimated ILS was then used for the retrieval process. An example of a cell measurement and a corresponding ILS retrieval is shown in Appendix A

$O_3$ was retrieved from the solar absorption spectra obtained with the photo-conductive MCT detector, in the $750\text{-}1300\,\mathrm{cm}^{-1}$

30 filter region at a spectral resolution of $0.005\,\mathrm{cm}^{-1}$ and applying an aperture of 1.7 mm. Every spectrum is calculated from averaging six scans with maximal resolution (total recording time of one spectrum is 12 minutes). Figure 2 gives an example of a typical measurement showing the five spectral windows that we use for the retrievals of $O_3$: $991.25\text{-}993.8\,\mathrm{cm}^{-1}$, $1001.47\text{-}1003.04\,\mathrm{cm}^{-1}$, $1005\text{-}1006\,\mathrm{cm}^{-1}$, $1007.3\text{-}1009\,\mathrm{cm}^{-1}$ and $1011.1\text{-}1013.6\,\mathrm{cm}^{-1}$. These are further refined windows if compared to the windows presented in Schneider and Hase (2008) (in order to reduce water vapour interferences) and are the standard

setting in PROFFIT NDACC/FTIR $O_3$ retrievals. We independently fit the $^{48}O_3$, the asymmetric $^{50}O_3$, the symmetric $^{50}O_3$ and all the $^{49}O_3$ isotopologues. All the $O_3$ isotopologues are fitted on a logarithmic volume mixing ratio scale. Furthermore, we perform simultaneous fits on a linear volume mixing ratio scale of the interfering species $H_2O$, $CO_2$ and $C_2H_4$, which all have small signatures in the used spectral windows. The position of the solar lines with respect to terrestrial lines is determined

in a proceeding analysis of a spectral window that contains well isolated solar and terrestrial lines.

Figure 3 characterises the $O_3$ remote sensing data obtained at Altzomoni. The left panel shows the averaging kernels and the right panels the estimated uncertainty for the $O_3$ retrieval made with the typical FTIR measurement at Altzomoni as shown in Fig. 2. This measurement gives a DOFS value of $4.26$, which is a typical value obtained for the Altzomoni $O_3$ retrievals. A DOFS of about 4.0 means that the $O_3$ values can be retrieved independently for about four different vertical altitude regions.

For illustration, we highlight the row kernels for the retrievals at 4, 17, 28 and 42 km altitude by thick lines and different colours. The retrieval at 4 km is mainly influenced by actual atmospheric $O_3$ variations between 4 and 10 km (black line), the retrieval at 17 km by actual variations between 15 and 23 km (red line), the retrieval at 28 km by actual variations between 23 and 32 km (green line) and the retrieval at 42 km by actual variations between 32 and 45 km (blue line).

The estimated errors are depicted in the right panels of Fig. 3. Shown are the square root values of the diagonal elements

of $\mathbf{S_e}$ calculated according to Eq. (6). Below the 30 km altitude total statistical and systematic errors are smaller than 5%. For higher altitudes the errors strongly increase and are larger than 10% above 40 km. Concerning the statistical error, uncertainties in temperature and baseline are the leading error sources, followed by uncertainties in the ILS and measurement noise. The systematic errors are dominated by uncertainties in the spectroscopic line parameters of $O_3$. Uncertainties in temperature, baseline and ILS are of secondary importance. Statistical and systematic errors due to uncertainties in the line of sight, solar

lines and interferences with atmospheric $H_2O$ variations are smaller than 0.1% throughout the atmosphere and are thus not depicted.

The errors for the total column abundances are listed in Table 2. They are calculated by applying the total column operator to the left and the transpose of the total column operator to the right of Eq. (6) and then taking the square root. The total column errors are between 2 and 3%, whereby the statistical error is mainly due to temperature uncertainty and the systematic error

due to uncertainties in spectroscopic line intensity parameters of $O_3$.

For this work we analysed 1672 individual Altzomoni spectra, measured on 143 individual days between November 2012 and February 2014. We found that occasionally there are spectra with rather high noise levels, which then lead to retrievals with relatively low DOFS values. The high noise levels are caused probably by clouds that pass through the line of sight when recording the interferograms. The Fourier transformation of such interferogram would then lead to artefacts in the spectrum,

like increased baseline offsets. In order to avoid that artefacts affect our study, we define as valid measurement only those for which the retrieval gives a DOFS value of at least 3.9, which is at the lower end of the DOFS values obtained for $O_3$ retrievals made with high resolution middle infrared solar absorption spectra (e.g. Schneider et al., 2008; García et al., 2012; Vigouroux et al., 2015). The DOFS filter leaves us with 1050 individual measurements made on 122 individual days between November 2012 and February 2014.

It is reasonable to assume that the Altzomoni solar absorption spectra are only very weakly affected by the large diurnal variations that take place in the polluted boundary layer. Most measurements are made before 14:00, when the boundary layer top altitude is below or weakly above the altitude of Altzomoni (Shaw et al., 2007; Baumgardner et al., 2009). Furthermore, very strong vertical mixing of air from below is generally linked to the presence of fog or clouds. Under such condition the

FTIR is not operated. Nevertheless, the $O_3$ background levels at Altzomoni depend on the outflow of polluted boundary layer air and on larger time scales (e.g. monthly means) the Altzomoni spectra are affected by the boundary layer pollution.

## 2.5 Observations at UNAM

The UNAM atmospheric observatory is located in the South of Mexico City on the roof of the CCA, within the main UNAM campus. The infrared solar absorption spectral measurements started in 2008 using a Bruker Opag 22 spectrometer (measuring

with spectral resolution of $0.5\,cm^{-1}$, Stremme et al., 2009). Since June 2010, spectra have been recorded with a Bruker Vertex 80 spectrometer allowing measurements with a maximum optical path difference of up to $12\,cm$, corresponding to a spectral resolution of $0.06\,cm^{-1}$. The spectrometer is equipped with a KBr beam splitter and two detectors (a nitrogen-cooled MCT detector and an InGaAs detector working at room temperature). In May 2012 we started to use four filters in the measurement routine, which cover different spectral sub-regions. The sunlight is guided by a solar tracker built in house which is covered

with a motorized dome. For details see Bezanilla et al. (2014).

The ILS of the interferometer is retrieved from HBr cell measurements (Hase et al., 1999; Hase, 2012) and then used for the retrieval process (an example of a cell measurement and a corresponding ILS retrieval is shown in Appendix A).

The spectra for the $O_3$ retrievals are measured with the photo-conductive MCT detector and with an aperture of $1.5\,mm$ using a long wave pass filter, which cuts on wavelength at $7.4\,\mu m$ (i.e. the filter is transparent for wavenumbers up to $1350\,cm^{-1}$).

Every spectrum is obtained from averaging ten scans with a resolution of $0.1\,cm^{-1}$, stored every 12 seconds in the computer. For the retrievals we use one spectral window between 991 and $1073\,cm^{-1}$. Figure 4 shows the spectral window for a measurement that has been taken in temporal coincidence with the Altzomoni measurement shown in Fig. 2. It is the same spectral window that has been used by Viatte et al. (2011). We fit the $O_3$ volume mixing ratios on a logarithmic volume mixing ratio scale, whereby all the different $O_3$ isotopologues are treated as a single species. As interfering species we simultaneously fit $H_2O$,

$CO_2$, $SO_2$, $NH_3$ and $C_2H_4$, whereby we only allow for scaling of the prescribed first guess profiles (which are the same as the profiles used as a priori, i.e. from the WACCM climatology). Furthermore, we fit phase and amplitude of channeling with a frequency of $0.39\,cm^{-1}$ (this kind of channeling is what we occasionally observe in the UNAM spectra). The solar lines simulated according to fixed model parameters.

Figure 5 shows the averaging kernels and error estimations for the retrieval made with the measurement as shown in Fig. 4.

The depicted row kernels belong to a averaging kernel matrix with a DOFS of 3.20, which is a typical value obtained for the UNAM $O_3$ retrievals and means that $O_3$ values can be retrieved independently for about three different vertical altitude regions. In Fig. 5 we highlight the row kernels corresponding to retrievals of the 2.3, 17 and $32\,km$ altitudes. The retrieval for $2.3\,km$ represents actual atmospheric $O_3$ variations between 2.3 and $10\,km$, the retrieval for $17\,km$ the actual variations between 15

and 24 km and the retrieval for 32 km the actual variations between 24 and 36 km. Even for spectra recorded with medium resolution the retrieval allows us to separate tropospheric, lower stratospheric and middle stratospheric $O_3$ variations.

Below 30 km the $O_3$ errors estimated for UNAM are higher than the errors estimated for Altzomoni. We calculate total statistical and systematic errors of about 7.5%. The respective statistical errors are mainly controlled by uncertainties in the
ILS, the baseline and the atmospheric temperatures. The respective systematic errors are dominated by uncertainties in the spectroscopic line parameters of $O_3$, the ILS and the baseline. Above 30 km the statistical errors increase to 10%, mainly caused by the increased importance of temperature uncertainties and measurement noise. The systematic errors slightly decrease to about 6% (above 30 km), mainly caused by decreasing importance of the uncertainties in the ILS and the baseline. Errors due to uncertainties of the line of sight, solar lines and interference with atmospheric $H_2O$ variations are smaller than 0.1%
throughout the atmosphere and not depicted.

The errors for the total column abundances are listed in Table 2. The respective UNAM errors are very similar to the Altzomoni errors (between 2 and 3%). As for Altzomoni the statistic error is mainly due to temperature uncertainty and the systematic error due to uncertainties in spectroscopic line intensity parameters of $O_3$.

For this work we analysed 1625 individual UNAM spectra measured on 88 individual days between November 2012 and
May 2013 (in June 2013 the measurements had to be interrupted due to construction works at UNAM). Similar to Altzomoni we filter the UNAM measurements with respect to the DOFS values obtained from the retrieval. We found that a DOFS value of 3.1 is a good threshold. Lower DOFS are strongly correlated to high noise levels and in particular to systematic residuals in the 1040-1045 $cm^{-1}$ region. This DOFS filter leaves us with 517 individual measurements made on 50 individual days between November 2012 and May 2013.

## 2.6 Logarithmic scale representation

In the equations and function (1)-(6) a logarithmic volume mixing ratio scale is used for the atmospheric $O_3$ state (we perform the $O_3$ retrieval on the logarithmic scale). In this context the errors as shown in Figs. 3 and 5 are from the logarithmic scale error covariance matrix $S_e$ according to Eq. (6), i.e. they represent actually errors in the logarithm of the $O_3$ volume mixing ratio. We interpret these logarithmic scale errors as relative errors, because $d\ln x = \frac{dx}{x}$ and $\Delta \ln x \approx \frac{\Delta x}{x}$.
Throughout the paper we will use differentials or differences in the logarithms of $O_3$ as equivalent to relative $O_3$ differentials or differences: like in this section in the context of error assessment studies and also in the following sections when comparing two different data sets. For the latter the differences in the logarithms of the two data sets will be calculated and then interpreted as relative differences.

## 3 Free tropospheric and stratospheric $O_3$

In this section we analyse the $O_3$ variations that are not linked to ozone smog events. To do so we work with free tropospheric $O_3$ data measured outside Mexico City and with stratospheric $O_3$ data. First we discuss the seasonal cycles as observed above

Altzomoni and then demonstrate to what extent the stratospheric $O_3$ variations can also be observed from the medium resolution instrument that is located in the Mexico City boundary layer.

## 3.1 Seasonal cycle above Altzomoni

Figure 6 depicts the monthly averages of $O_3$ total column amounts (top panel) and volume mixing ratios for different altitudes
(bottom panel) for the months where there are at least Altzomini FTIR measurements on three different days. The error bars indicate the standard error of the mean, which is the standard deviation divided by the square root of the number of measurements used when calculating the mean value. The volume mixing ratios are presented for the altitudes whose row kernels are highlighted in the left panel of Fig. 3. The seasonal variation at the different altitudes is shown in a single graph, but with different volume mixing ratio scale. The data points and the corresponding volume mixing ratio scales can be identified
by different colours: dark gray for 4 km, red for 17 km, green for 28 km and blue for 42 km.

The seasonal cycle of the $O_3$ total column amounts above Altzomoni is rather smooth with a maximum in summer (in August 2013 almost 290 DU is reached) and a minimum in January (the January average is 241 DU). This is significantly different to the seasonal cycles of $O_3$ total column amounts as observed over subtropical, mid-latitudinal and polar sites. There the maximum is reached in spring and the minimum in autumn (e.g. Vigouroux et al., 2015). A look on the seasonal cycles for
different altitudes can help to understand the particularity of the seasonal $O_3$ distributions over central Mexico.

In the free troposphere (represented by the 4 km retrievals) there seems to be two clearly distinguishable periods. A winter period from November to February with $O_3$ values of about 0.04 ppmv and a second period from April to August with $O_3$ values between 0.05 and 0.06 ppmv. This is similar to the seasonal cycle of mid-latitudinal and subtropical tropospheric $O_3$ found for the two high altitude stations Izaña and Jungfraujoch (García et al., 2012; Vigouroux et al., 2015). In an exemplary
study for 2006, Thompson et al. (2008) found that most of this spring-to-summer increase over central America occurs between 5 and 12 km altitude. They attributed the increased tropospheric background $O_3$ levels to accumulation of earlier stratospheric $O_3$ input and to more $O_3$ pollution being imported from lower altitudes. The importance of the Mexico City emissions on free tropospheric $O_3$ levels in the surroundings of Mexico City has also been demonstrated in the context of the model study of Emmons et al. (2010).

In the upper troposphere and lower stratosphere (UTLS, represented by the 17 km retrievals) the seasonal variation is smoother and there is a slow but consistent gradual increase of $O_3$ concentrations between January and August. Unfortunately we have not a significant number of observations in September and October, but the August, November and December data seem to suggest a gradual decrease between August and December/January. The maximum in late summer and the minimum in winter is very different to what has been observed at other sites (Vigouroux et al., 2015), where the seasonal cycle of the UTLS
is strongly linked to the seasonal variation of the tropopause height. Between the subtropics and polar regions the tropopause is highest at the end of the summer and lowest in winter/spring, resulting in low UTLS $O_3$ volume mixing ratios in summer and high ratios in winter. Above Altzomoni it is the other way round. The reason is the importance of isentropic mixing of $O_3$ rich air from higher latitudes into the UTLS above Altzomoni. This mixing is strongest in summer and is also seen in space-based observations (Stolarski et al., 2014). Shuckburgh et al. (2009) investigated the longitudinal, seasonal and interannual variations

of these mixing events and showed that the mixing is strongest in the northern hemispheric summer and close to the Asian and North American monsoon regions. Furthermore, they found in particular strong mixing for negative ENSO (El Niño Southern Oscillation) years. Isentropic mixing may be determining the seasonal signal in the UTLS region also affecting the seasonal cycle of the total column amounts in central Mexico, but this needs to be further investigated. Long-term FTIR measurements at Altzomoni, which is situated in the North American monsoon region, will allow to study these stratosphere-troposphere exchange processes and their link to climate patterns like ENSO.

In the middle stratosphere (represented by the 28 km retrievals) we observe an increase between January and May and a decrease between May and November, i.e. the maximum values are already achieved in May. For the retrieval at 42 km (representative for the middle and upper stratosphere), we observe a maximum in late summer and a minimum in winter. These middle and upper stratospheric cycles at Altzomoni are similar to the cycles as observed at the subtropical site of Izaña (García et al., 2012).

### 3.2  Intercomparison between UNAM and Altzomoni data

When comparing different remote sensing products we have to consider the respective averaging kernels. The kernels as depicted in Figs. 3 and 5 reveal that at Altzomoni we can observe more details of the vertical $O_3$ distribution as compared to UNAM. Therefore before comparing the Altzomoni and the UNAM retrieval products we have to account for this difference by smoothing the retrieved Altzomoni $O_3$ state vector ($x_{\mathrm{ALTZ}}$) with the UNAM averaging kernel ($\mathbf{A}_{\mathrm{UNAM}}$).

$$x^*_{\mathrm{ALTZ}} = \mathbf{A}_{\mathrm{UNAM}}(x_{\mathrm{ALTZ}} - x_{a\,\mathrm{ALTZ}}) + x_{a\,\mathrm{ALTZ}}. \tag{7}$$

Here the vectors $x_{\mathrm{ALTZ}}$ and $x_{a\,\mathrm{ALTZ}}$ have been expanded by three additional dimensions, which correspond to the three altitude grid levels below 4 km. For these three grid levels we set $x_{\mathrm{ALTZ}} = x_{a\,\mathrm{ALTZ}} = x_{a\,\mathrm{UNAM}}$.

Applying Eq. (7) makes the data better comparable but does not fully remove the effect of the different averaging kernels. In order to assure that we perform a reasonable comparison we calculate the covariances $\mathbf{S_{cmp}}$ that estimates the averaging kernel induced uncertainty for the comparison between the UNAM remote sensing data and the Altzomoni remote sensing data after smoothing according to Eq. (7):

$$\mathbf{S_{cmp}} = (\mathbf{A}_{\mathrm{UNAM}} - \mathbf{A}_{\mathrm{UNAM}}\mathbf{A}_{\mathrm{ALTZ}})\mathbf{S_{cov}}(\mathbf{A}_{\mathrm{UNAM}} - \mathbf{A}_{\mathrm{UNAM}}\mathbf{A}_{\mathrm{ALTZ}})^T. \tag{8}$$

whereby $\mathbf{A}_{\mathrm{UNAM}}$ is the averaging kernel matrix for UNAM and $\mathbf{A}_{\mathrm{ALTZ}}$ the averaging kernel matrix for Altzomoni, being expanded by three columns and three rows with 0.0 entries corresponding to the three altitude grids below 4 km altitude. The matrix $\mathbf{S_{cov}}$ describes the $O_3$ covariances in the atmosphere. Here we use $\mathbf{S_{cov}} = \mathbf{S_{cov,meas.}}$ (see Appendix B for more details). For a reasonable comparison we require that the square root of the diagonal of $\mathbf{S_{cmp}}$ that represents the altitude under consideration is smaller than 5%.

Figure 7 shows a comparison the total column amounts as well as for volume mixing ratios at 17 km and 32 km a.s.l. after applying the smoothing from Eq. (7) and the aforementioned filtering. We pair UNAM and Altzomoni data that are measured within 2h and then calculate the hourly means for all the pairs. This gives 56 individual data pairs that belong to measurements made on 23 individual days and during 6 different months from November 2012 to April 2013.

The total columns are calculated from the retrieved UNAM profiles (state vector $x_{\mathrm{UNAM}}$) and the retrieved and smoothed Altzomoni profiles (state vector $x^*_{\mathrm{ALTZ}}$ above 4 km, according to Eq. 7). As documented by the left graph of Fig. 1 according to the model WACCM about 97% of the $O_3$ total column abundances as measured at UNAM are situated above the altitude of Altzomoni and indeed we observe a very good correlation between the data from both stations (a correlation coefficient $R^2$ of 91% and a slope $m$ for the linear regression line of 0.99). The mean difference (UNAM − Altzomoni) is +4.8% and the $1\sigma$ scatter is 1.5%. The bias of +4.8% is in reasonable agreement to the typical relative $O_3$ abundances between 2.3 and 4 km (see black line in the left panel of Fig. 1).

The correlations between the volume mixing ratios obtained at 17 km are also strong ($R^2$ of 87%). However the slope of the regression line is larger than unity ($m = 1.14$). For the mixing ratios at 32 km the correlation is a bit weaker, although still clearly observable ($R^2$ is 75%), and the slope of the regression line is close to unity ($m = 0.94$). The mean differences and scatter (UNAM − Altzomoni) are +13.2%±4.4% for 17 km and −1.2%±2.4% for 32 km, respectively. The bias and scatter between the two data sets might be explained by uncorrelated errors in the data, remaining differences in the smoothing characteristics (although of less importance due to applying the filter based on the calculation according to Eq. 8) and the detection of different air masses.

## 4   Boundary layer

This section focuses on the $O_3$ volume mixing ratios in the boundary layer of the Mexico City basins, whose variations are driven mainly by photochemistry from anthropogenic pollutants (photochemical ozone smog).

### 4.1   In-situ monitoring of air pollution in Mexico City

For validating the boundary layer remote sensing product we use in-situ measurements made in Mexico City in the framework of the RAMA (Red Automática de Monitoreo Atmosférico) network. RAMA performs continuous measurements of different gases and particles in 34 stations spread around Mexico City in order to give information about air quality of this megacity (more details about RAMA are available at www.aire.df.gob.mx/default.php).

The in-situ $O_3$ monitoring data are freely available with an hourly time resolution. We work with the three in-situ stations Pedregal (PED), Santa Ursula (SUR) and Coyoacan (COY), which are all situated within a circle of about 5 km radius around the UNAM station, thereby constituting an excellent reference for assessing the potential of the UNAM remote sensing experiment for observing boundary layer $O_3$ mixing ratios. We calculate the mean value from the three stations for each hour and only consider situations where all three stations provide data.

Figure 8 depicts the monthly means of the diurnal cycles obtained from these three in-situ stations and represents the known $O_3$ variability with highest monthly surface averages during the March-May months. It is during this period that the city suffers the largest number of exceedances and that the government activates the contingency plans for minimizing the high $O_3$ pollution episodes. The observed high diurnal variability, which is evidently larger than the seasonal variability, reveals the photochemical nature and reactivity of this urban atmosphere. The blue dots depict the time period when the FTIR

measurements are typically performed (between 11 and 14 local time). Within these three hours $O_3$ concentrations show a strong increase. Any comparison study has to consider these fast changing $O_3$ concentrations and we only compare data that are measured within the same hour. A discussion on $O_3$ boundary layer variability of Mexico City on time scales beyond the diurnal time scale is given in Barrett and Raga (2016).

## 4.2  Intercomparison of remote sensing data with in-situ data

For the comparison between the in-situ and remote sensing data, we require that during a certain hour (given in local time) all the three experiments provide at least one measurement. We created a table that contains the data from the three experiments for these coincidences and then calculated the hourly mean data. This gives 33 individual hourly mean data triplets belonging to measurements made during 16 individual days during 6 different months (between November 2012 and April 2013).

Figure 9 shows the correlation plots between the $O_3$ in-situ data (measured in the Mexico City boundary layer at about 2.3 km a.s.l.) and the $O_3$ values obtained for the lowermost altitude for the retrievals at Altzomoni and UNAM. The left panel depicts the correlation between the in-situ data and the Altzomoni retrieval product for 4 km a.s.l.. We observe no correlation ($R^2 < 1\%$) and a mean difference and $1\sigma$ standard deviation of the difference of -37.6%±25.8% (for Altzomoni − In-situ). This is not surprising given the significant horizontal distance between the location of the in-situ instruments and Altzomoni and the fact the Altzomoni data have no sensitivity below 4 km a.s.l.. This implies that the variation as observed by the in-situ instruments are rather local and vertically and/or horizontally limited to the area around the in-situ instruments (or the basin of Mexico City).

The middle panel of Fig. 9 presents the correlation between the same in-situ data and the UNAM retrieval product for 2.3 km a.s.l.. There is a clear correlation between the two data sets ($R^2 = 54\%$), although the UNAM FTIR data do not fully capture the magnitude of the $O_3$ variability at the surface (slope $m$ of the linear regression line is only 0.53). A slope of below 0.5 is actually what can be expected from the respective averaging kernels. The thick black line in Fig. 10 depicts the row kernel for the UNAM retrieval at 2.3 km. Summing up all the contributions of the row kernel for the boundary layer (values between surface and 4 km a.s.l.) we get about 0.35. If we assume that the diurnal $O_3$ increase is present throughout the entire boundary layer and if we further assume that it is stronger a few hundred meters above the surface (compared to the increase as observed in the mean values of the three in-situ stations), we can expect a slope above 0.35. Between 9:00 and 14:00 local time such assumption is reasonable, judging from the vertical $O_3$ profiles shown in Velasco et al. (2008) and the depth of the boundary layer (Shaw et al., 2007). The mean difference and $1\sigma$ standard deviation of the difference is -11.2%±16.1% (for UNAM − In-situ)

The good agreement with the in-situ data empirically proves the profiling capability of the UNAM FTIR experiment: it is able to detect the $O_3$ variations that take place in the Mexico City boundary layer, although it is a relatively thin layer containing only a small portion of the total column $O_3$ abundance above UNAM (see left panel in Fig. 1).

## 4.3 A combined UNAM/Altzomoni remote sensing product for the boundary layer

We have two remote sensing experiments located close to each other but at different altitudes. In this section we present a product that combines the measurement made by the two instruments. The objective is to investigate if the combination can improve the remote sensing boundary layer data.

A simple method is to calculate the differences between the total columns measured above UNAM and above Altzomoni, i.e. the differences of the total columns values as depicted in the left panel of Fig. 7. However, we have to consider that the partial column between 2.3 km and 4 km is only about 2.5% of the total column amount (see left panel of Fig. 1). Even after treating the Altzomoni data with the UNAM kernel (according to Eq. 7), the UNAM and the Altzomoni total column have still not the same sensitivity. Actually if we use the metric as described in the context of Eq. (8) we find that the partial columns above 4 km altitude can typically agree only within 0.9%. A better agreement cannot be expected due to the different sensitivities of the two remote sensing experiments. In addition there are errors in the total column amounts, which are estimated to 2.5-3% for each of the two experiments (see Table 2). In summary, the difference between the two column amounts is affected by different sensitivities and errors, whereby both effects together sum up to about 4-5%. This is larger than the expected value for the difference of about 2.5% and in addition it has to be consider that the boundary layer sensitivity is only 35% (sum of the diagonal elements of the averaging kernel corresponding to boundary layer altitudes, see discussion in the context of Fig. 9, middle panel). We cannot expect detecting a 2.5%×35%<0.9% signal by a measure that has an uncertainty of 4-5% and in consequence the difference of the total column amounts cannot give useful information about the partial column between 2.3 km and 4 km. Actually, we have calculated the total column amount differences and find indeed no correlation to the boundary layer in-situ data.

We need a more sophisticated method for combining the two experiments. We can start with the UNAM product at 2.3 km, which shows correlations to the boundary layer in-situ data (recall middle panel of Fig. 9). The typical row kernel for the UNAM retrieval at 2.3 km is depicted as thick black line in Fig. 10. The kernel indicates some sensitivity for altitudes at and below 4 km, although the sum of the diagonal elements of the kernel corresponding to these altitudes is typically 0.35, i.e. significantly smaller than 1.0. It can also be seen that the 2.3 km retrieval is sensitive to the actual atmosphere above 4 km (the row kernel values only slowly decrease for altitudes above 4 km), i.e. free tropospheric $O_3$ variations can significantly interfere with the boundary layer variations. Our idea is to improve the sensitivity for the boundary layer and at the same time reduce the interferences from higher altitudes by combining the UNAM measurements with the Altzomoni measurements. The Altzomoni data are promising for reducing the interferences because it is sensitive to the free troposphere above 4 km, but completely insensitive to the boundary layer below 4 km (see the row averaging kernel for the Altzomoni retrieval at 4 km depicted as thick grey line in Fig. 10).

### 4.3.1 Analytical description of the combined product

We introduce an operator $\mathbf{C}$ for combining the two retrievals:

$$\mathbf{C} = \frac{1}{\sum\limits_{i\,\epsilon\,\mathrm{BL}} a_{\mathrm{comb}}(i,i)} \left( \begin{array}{cc} \mathbb{I} & -\mathbf{A}_{\mathrm{UNAM}} \end{array} \right). \tag{9}$$

Here $\mathbb{I}$ is a $nol \times nol$ identity matrix and $\mathbf{A}_{\mathrm{UNAM}}$ the $nol \times nol$ averaging kernel matrix for UNAM, whereby $nol$ is the number
of grid points of the model atmosphere used for the UNAM retrieval process (i.e. here $nol = 44$). So $\mathbf{C}$ is a $nol \times (2 \times nol)$
matrix. The matrix is normalised to $\sum_{i\epsilon\mathrm{BL}} a_{\mathrm{comb}}(i,i)$, where $a_{\mathrm{comb}}(i,i)$ are the diagonal elements of $\mathbf{A}_{\mathrm{comb}}$ and $i\epsilon\mathrm{BL}$ are the
indices for the altitudes at and below 4 km (i.e. the boundary layer, BL). The matrix $\mathbf{A}_{\mathrm{comb}}$ is calculated according to Eq. (11),
so for calculating $\mathbf{A}_{\mathrm{comb}}$ we actually need the operator $\mathbf{C}$, which in its turn needs diagonal elements from matrix $\mathbf{A}_{\mathrm{comb}}$. So
we need two steps to calculate $\mathbf{C}$ and $\mathbf{A}_{\mathrm{comb}}$ correctly: first we apply Eq. (9) using 1.0 for the normalisation, then we apply
$\mathbf{A}_{\mathrm{comb}}$ according to Eq. (11). This gives us the correct normalisation factor for calculating the correct operator $\mathbf{C}$.

With the combination operator $\mathbf{C}$ the combined product can be calculated and comprehensively characterised in a similar
way as the individual products. The combined state vector $\boldsymbol{x}_{\mathrm{comb}}$ can be calculated from the UNAM and Altzomoni state
vectors ($\boldsymbol{x}_{\mathrm{UNAM}}$ and $\boldsymbol{x}_{\mathrm{ALTZ}}$, respectively) as follows:

$$\boldsymbol{x}_{\mathrm{comb}} = \mathbf{C} \left[ \left( \begin{array}{c} \boldsymbol{x}_{\mathrm{UNAM}} \\ \boldsymbol{x}_{\mathrm{ALTZ}} \end{array} \right) - \left( \begin{array}{c} \boldsymbol{x}_{\boldsymbol{a}\,\mathrm{UNAM}} \\ \boldsymbol{x}_{\boldsymbol{a}\,\mathrm{ALTZ}} \end{array} \right) \right] + \boldsymbol{x}_{\boldsymbol{a}\,\mathrm{UNAM}}. \tag{10}$$

Here $\boldsymbol{x}_{\boldsymbol{a}\,\mathrm{UNAM}}$ and $\boldsymbol{x}_{\boldsymbol{a}\,\mathrm{ALTZ}}$ are the a priori state vectors for UNAM and Altzomoni, which are identical. As for Eq. (7) the
Altzomoni state vectors are expanded to 44 dimensions and we define $\boldsymbol{x}_{\mathrm{ALTZ}} = \boldsymbol{x}_{\boldsymbol{a}\,\mathrm{ALTZ}} = \boldsymbol{x}_{\boldsymbol{a}\,\mathrm{UNAM}}$ for the three vector
components corresponding to the three altitude levels below 4 km.

The averaging kernel for the combined product can be calculated as:

$$\mathbf{A}_{\mathrm{comb}} = \mathbf{C} \left( \begin{array}{c} \mathbf{A}_{\mathrm{UNAM}} \\ \mathbf{A}_{\mathrm{ALTZ}} \end{array} \right). \tag{11}$$

Here the matrix $\mathbf{A}_{\mathrm{ALTZ}}$ is the same as in Eq. (8), i.e. the original averaging kernel matrix of Altzomoni expanded by three
columns and three rows with 0.0 entries corresponding to the three altitude grids below 4 km altitude (this gives a $nol \times nol$
matrix).

Similarly to Eq. (6) we can calculate error covariances for the combined product:

$$\mathbf{S}_{\mathbf{e},\mathrm{comb}} = \mathbf{C} \left( \begin{array}{cc} \mathbf{G}_{\mathrm{UNAM}} & 0 \\ 0 & \mathbf{G}_{\mathrm{ALTZ}} \end{array} \right) \left( \begin{array}{cc} \mathbf{K}_{\mathbf{p},\mathrm{UNAM}} & 0 \\ 0 & \mathbf{K}_{\mathbf{p},\mathrm{ALTZ}} \end{array} \right) \left( \begin{array}{cc} \mathbf{S}_{\mathbf{p},\mathrm{UNAM}} & \mathbf{S}_{\mathbf{p},\mathrm{x}} \\ \mathbf{S}_{\mathbf{p},\mathrm{x}} & \mathbf{S}_{\mathbf{p},\mathrm{ALTZ}} \end{array} \right)$$

$$\left( \begin{array}{cc} \mathbf{K}_{\mathbf{p},\mathrm{UNAM}}{}^T & 0 \\ 0 & \mathbf{K}_{\mathbf{p},\mathrm{ALTZ}}{}^T \end{array} \right) \left( \begin{array}{cc} \mathbf{G}_{\mathrm{UNAM}}{}^T & 0 \\ 0 & \mathbf{G}_{\mathrm{ALTZ}}{}^T \end{array} \right) \mathbf{C}^T. \tag{12}$$

Here the matrix $\mathbf{G}_{\mathrm{ALTZ}}$ is the original gain matrix for Altzomoni expanded by three lines (corresponding to the three altitude
grid levels below 4 km) to $nol$ lines. The entries in these lines are 0.0. The matrices $\mathbf{K}_{\mathbf{p},\mathrm{UNAM}}$ and $\mathbf{K}_{\mathbf{p},\mathrm{ALTZ}}$ are the Jacobians

with respect to parameter $p$ and the matrices $\mathbf{S}_{\mathbf{p},\text{UNAM}}$ and $\mathbf{S}_{\mathbf{p},\text{ALTZ}}$ give the the uncertainty covariances for parameters $p$ for the UNAM and the Altzomoni retrievals, respectively. The block $\mathbf{S}_{\mathbf{p},\text{x}}$ define the correlation between uncertainties for UNAM and Altzomoni. Here we assume that all uncertainties are uncorrelated, except for the temperature uncertainties above 12.5 km, for which we assume full correlation (i.e. $\mathbf{S}_{\mathbf{p},\text{x}}$ has only entries for the elements that correspond to temperature uncertainties above 12.5 km). In the troposphere we assume that the uncertainty in the temperatures above UNAM and Altzomoni are uncorrelated, because here small scale variation are likely. For altitudes above 12.5 km we assume correlated temperature uncertainties because at these altitudes the variations on smaller scales are less likely. In this context please be aware that for both sites we assume the same temperatures, which are from NCEP and climatology of CIRA for altitudes above 50 km.

### 4.3.2 The working principle of the combined product

In the previous subsection we give the formula that analytically describe the characteristics the combined product. It is fully traceable back to the Jacobians matrices, gain matrices and averaging kernel matrices of the individual retrieval products (see Eqs. 10 - 12). In this subsection we provide some additional intuitive explanations with the objective of better communicating the working principle of the method.

For our combined product we are able to optimize the boundary layer data quality, because we add to the measurement in the boundary layer (the UNAM spectrum) a second measurement above the boundary layer (the Altzomoni spectrum). The best way to do that would be to fit both measured spectra within a single inversion process (similar to what is done for limb sounding retrievals, e.g. Fischer et al., 2008), which would, however, mean the setup of a new inversion algorithm software. Our method is a work-around of such multi-spectral inversion algorithm, because it works with the two individual retrieval results and not with the individual measurements. The method consists in an a posteriori combination the two individual retrieval results with the objective of optimally exploiting the synergies of the two individual measurements.

Equation (10) shows how the combined product is calculated. By inserting $\mathbf{C}$ from Eq. (9) we get:

$$\frac{1}{\sum\limits_{i\,\epsilon\,\text{BL}} a_{\text{comb}}(i,i)}[(\boldsymbol{x}_{\text{UNAM}} - \boldsymbol{x_a}_{\text{UNAM}}) - \mathbf{A}_{\text{UNAM}}(\boldsymbol{x}_{\text{ALTZ}} - \boldsymbol{x_a}_{\text{ALTZ}})] + \boldsymbol{x_a}_{\text{UNAM}}. \tag{13}$$

This reveals that the method consists in principle in calculating the differences of the two measurements, but also accounts for the different sensitivities and assures that the kernel of the combined product is normalized for the boundary layer (it is required that the sum of its boundary layer diagonal elements is 1.0).

### 4.3.3 Discussion

The row of $\mathbf{A}_{\text{comb}}$ corresponding to 2.3 km is depicted as thick red line in Fig. 10 and indicates that, in comparison to the UNAM product, the combined retrieval product has much larger sensitivity in the boundary layer but at the same time much less sensitivity to variations that occur above 4 km. In this context our approach using two remote sensing experiments observing at different altitudes seems to be very promising.

However, we also have to consider the errors. By combining two measurements we increase the errors because the errors of the two measurements are largely independent. In addition, by using the normalisation factor we make the data more sensitive

to actual atmospheric variations, but also to uncertainty sources. Table 3 collects the errors estimated for the UNAM and the combined product at 2.3 km altitude. The values are the square roots of the diagonal entries of $\mathbf{S_e}$ and $\mathbf{S_{e,comb}}$, according to Eqs. (6) and (12), respectively, that correspond to the altitude of 2.3 km. Because we assume that the UNAM and Altzomoni uncertainties are uncorrelated (except for temperatures uncertainties above 12.5 km) and because we increase the sensitivity

(by using the normalisation factor), the errors are significantly larger in the combined product than in the UNAM product. We estimate that the total statistical and systematic errors can be as large as 30%. So while the combination strongly reduces interferences from $O_3$ variations above 4 km and increases the sensitivity to actual boundary layer $O_3$ variations, it significantly increases the errors by summing up error contributions from two different experiments and by increasing the sensitivity to all uncertainty sources.

In the right panel of Fig. 9 we compare the combined product with the in-situ data (for exactly the same coincidences that are shown in the other panels). We find that the combined product clearly correlates better with the in-situ data than the UNAM product ($R^2$ increases from 54% to 68%). The slope $m$ does increase significantly, which is achieved by the normalisation factor when calculating $\mathbf{C}$ according to Eq. (9). The normalisation factor assures that for the combined product the DOFS for the boundary layer is 1.0. This result is a clear empirical evidence that by combining the solar absorption spectra measurements

of UNAM and Altzomoni according to Eq. (10), we can generate data that well capture the variations taking place in the Mexico City boundary layer. However, at the same time we observe a significant bias with respect to the in-situ data. The mean difference and $1\sigma$ standard deviation of the difference is +30.4%±19.0% (Combined − In-situ). This bias is above the upper limit of the estimated systematic errors.

There can be different reasons for this bias. It might be that we underestimate the systematic errors by assuming too low

uncertainties in the spectroscopic line parameter data of $O_3$ or in the modulation efficiency. On the other hand at least part of the systematic difference between the in-situ and the remote sensing data might be explained by the fact that the two measurement techniques observe different air masses. The in-situ instruments detect air at 2-5 m above the surface, while the remote sensing data represent the first 2 km above the surface. In model studies (e.g. Raga and Raga, 2000) as well as from observations (e.g. Velasco et al., 2008; Thompson et al., 2008), increased $O_3$ concentrations have been found about 1 km above the surface. This

increase from the surface up to 1 km above the surface seems to be particularly important in the morning and disappears at midday (Velasco et al., 2008). So when interpreting the differences between the in-situ and remote sensing data we have to keep in mind that both data represent different altitude regions. Actually this makes the remote sensing data especially interesting: they complement the in-situ data by giving information about a 2 km thick layer above the surface, and thus opening the possibility to detect extraordinary enhancements such as in Bezanilla et al. (2014), or the possibility to quantify a residual

layer within the boundary layer as is suggested in this study.

## 5   Summary and outlook

To our knowledge we present the first ground-based FTIR remote sensing data of $O_3$ profiles for Latin America. We work with two different FTIR experiments located one in the megacity of Mexico City and another about 1700 m above the city at a

distance of 60 km. The instrument in the city is a moderate resolution instrument and is installed on the top of a building at the main UNAM campus. The instrument outside the city is a high resolution instrument, situated on the high altitude observatory of Altzomoni and contributes to the NDACC (currently it is the highest NDACC FTIR station worldwide).

It is demonstrated that with the high altitude NDACC instrument one can typically detect the $O_3$ volume mixing ratio variations of four different altitude regions: the free troposphere, the UTLS, the middle stratosphere and the middle/upper stratosphere. This is in agreement to other sites for which ground-based FTIR remote sensing profiles of $O_3$ have been characterised. We found that the error in the profiles are typically within 5% except for altitudes above 30 km, where they can reach 10-20%. Statistic errors are dominated by uncertainties in the used atmospheric temperature profiles and systematic errors are mainly due to uncertainties in the spectroscopic line parameters of $O_3$. These results are in agreement to previous error estimations studies for NDACC FTIR $O_3$ data retrievals at other stations.

We show a first error estimation for $O_3$ profiles obtained for retrievals that use moderate resolution solar absorption spectra (the UNAM measurements). The estimated profile errors are below 10% between surface and 50 km, being dominated by uncertainties in atmospheric temperature (statistical error) and uncertainties baseline of the measured spectra and the spectroscopic line parameters of $O_3$ (systematic error). It is documented that remote sensing with the moderate resolution instruments still allows for detecting three different altitude regions: the first 7 km above the surface, the UTLS region and the middle/upper stratosphere.

The theoretical error estimations are confirmed by comparing the UNAM FTIR product with the Altzomoni FTIR product. We find very good agreement for both data sets concerning total column amounts, UTLS volume mixing ratios and the ratios in the middle/upper stratosphere. A comparison of the lowermost layer is not possible, because the UNAM FTIR data are strongly affected by local $O_3$ variations in the polluted boundary layer of Mexico City, whereas the Altzomoni FTIR experiment is measuring mainly above this layer. For empirically validating the boundary layer data obtained from the UNAM remote sensing retrievals we use in-situ data obtained at three stations close to the UNAM instrument. We found a good correlation between the UNAM FTIR and the in-situ boundary layer $O_3$ data and a slope of the linear regression line that is in agreement with the sensitivity as given by the averaging kernels.

The Altzomoni FTIR measurements will become very useful in the future, when several years of measurements will be available. The seasonal cycles observed in the total column and the UTLS above Altzomoni are distinct to the cycles as observed at all the other NDACC FTIR stations in the subtropics, mid-latitudes and polar regions. The observation of a UTLS $O_3$ maximum in late summer instead of late winter and spring (like at other sites) is most probably due to isentropic mixing of $O_3$ rich air from the high latitudinal stratosphere. This mixing takes place in the northern tropics. In previous studies it has been shown that it is linked to climate patterns like the Asian and North American monsoons and that it is affected by the ENSO phase. Establishing the Altzomoni FTIR as a long-term activity might in the future enable us to investigate the importance of climate patterns, like the North American monsoon or the ENSO, for such stratosphere-troposphere exchange events.

Having two nearby remote sensing measurements, a first one in the boundary layer and a second one just above this layer most of the time, gives the opportunity of creating a combined remote sensing product with improved representativeness of the near surface processes. We have introduced the theory of such combined product and characterise it by calculating averaging

kernels and errors. We are able to prove that the combined FTIR $O_3$ boundary layer product better represents the near surface variations than the product obtained by a single FTIR experiment. The approach for combining the retrievals of the two FTIR experiments can be applied for any other trace gas in order to reliably detect the Mexico City boundary layer variations of many different trace gases. Nevertheless, for a scientific usage of this remote sensing boundary layer product it would be

important to better understand the bias of 30% with respect to the in-situ data. The bias can be due to systematic errors in the remote sensing data, but also due to the fact that the remote sensing experiments observe different air masses. By measuring $O_3$ profiles with radiosondes one can detect the same air masses as the remote sensing experiments. In this context, performing in-situ $O_3$ profile measurements at the UNAM station (radiosonde or balloon) in coincidence to Altzomoni and UNAM FTIR measurements would be very helpful, since it could help understanding the actual systematic error in the boundary layer remote

sensing data. With more data and further analysis, our product could be used to quantify the $O_3$ residual layer and to understand the complex dynamical processes affecting the large variability and high episodes taking place at the surface.

## 6    Data availability

NDACC FTIR $O_3$ data of Altzomoni have been stored on the NDACC database: ftp://ftp.cpc.ncep.noaa.gov/ndacc/station/altzomoni/hdf/ftir/. However, please be aware that the data on the NDACC database have been retrieved using a single microwindow at 1000-

1005 cm$^{-1}$ (the NDACC standard setting), which is different to the retrieval setting used in this study. In consequence the Altzomoni $O_3$ data on the NDACC database are slightly different from the here presented $O_3$ data. The here presented Altzomoni and UNAM $O_3$ data are so far only available by request to the authors. The RAMA data are available at: www.aire.df.gob.mx/default.php.

## Appendix A:  Measurements of the instrumental line shape

This appendix presents instrumental line shapes (ILS) for the two instruments as retrieved from measurements with an HBr

cell (Hase, 2012) using the LINEFIT software (Hase et al., 1999).

## A1    Bruker IFS 120/5 HR at Altzomoni

At Altzomoni cell measurements have been performed several times per year. Figure 11 shows a cell measurement taken in November 2013 and its analysis. The left panel depicts a spectral window with an HBr absorption line, the central panel the retrieved modulation efficiency and the right panel the averaging kernels for the modulation efficiency retrieval. We observe a

decay in modulation efficiency of 8% (modulation efficiency of 0.92 at 120 cm optical path difference, OPD). The averaging kernels indicate that the retrieval is very sensitive up to an OPD of 120 cm. At longer OPDs the sensitivity starts to decrease, i.e the values obtained for an OPD of 180 cm have an increased uncertainty.

## A2 Bruker Vertex 80 at UNAM

At UNAM cell measurements have been made every three years. For the future we plan to reduce the time periods between these measurements. Figure 12 shows the cell measurement taken in September 2011 and its analysis. The panels from the left to the right show an analysed spectral window with an HBr absorption signature, the retrieved modulation efficiency and the averaging kernels for the modulation efficiency retrieval. There is a decay of the modulation efficiency of about 9% at OPDs above 6 cm. It is also important to have a look on the averaging kernels. It seems that the modulation efficiency retrieved above 7 cm OPD is mainly affected by the real OPD below 7 cm. In consequence the modulation efficiency obtained for 9 cm OPD has an increased uncertainty.

## Appendix B: The smoothing characteristic of remote sensing data

A very important characteristic of the remote sensing product is its limited sensitivity and vertical resolution. This characteristic is documented by the averaging kernels (see Figs. 3 and 5). The kernels are essential for a scientific application of the remote sensing data: for model studies the kernels are applied to the model data, only then the remote sensing data can serve as a reference (e.g. Langerock et al., 2015). Similarly, profile measurements that offer high vertical resolutions are convolved by the kernels and can then be compared to the remote sensing data (e.g. Schneider et al., 2008).

In several scientific studies this limited sensitivity and vertical resolution is treated as an error component, following the "smoothing error" concept of Rodgers (2000), i.e. calculating a "smoothing error" covariance according to $(\mathbf{A} - \mathbf{I})\mathbf{S_{cov}}(\mathbf{A} - \mathbf{I})^T$, where $\mathbf{A}$ is the averaging kernel, $\mathbf{I}$ the identity matrix and $\mathbf{S_{cov}}$ the assumed atmospheric covariance. The so-obtained error covariances very strongly depend on the atmospheric covariances $\mathbf{S_{cov}}$ to which they refer to. For instance, if we are interested in the uncertainty of comparing an ensemble of model data with coinciding remote sensing data, we need to calculate the atmospheric covariances from exactly the ensemble of model data that coincides with the remote sensing data. These "smoothing error" covariances are then only valid for this specific situation and they are not valid for a comparison to a smaller or larger model data ensemble (e.g. covering a different time period), for comparison to other models with a different temporal and spatial resolution, or for a comparison to other measurements (which might have a much better temporal and spatial resolution than the model). Furthermore, the "smoothing error" covariances are specific for the vertical gridding used for the remote sensing retrievals. A critical discussion of the "smoothing error" concept is given in von Clarmann (2014).

In the following Subsections we investigate atmospheric $O_3$ covariances obtained from model and measurement data. Then we provide some calculation that are linked to the "smoothing error" concept and also briefly discuss their validity.

### B1 Atmospheric covariances

The covariance matrix $\mathbf{S_{cov}}$ can be written as $\mathbf{S_{cov}} = \mathbf{\Sigma}\mathbf{\Gamma}\mathbf{\Sigma}^T$, where $\mathbf{\Sigma}$ is a diagonal matrix containing the $1\sigma$ variability at the different altitude levels and $\mathbf{\Gamma}$ the correlation matrix documenting how the variations between the different altitudes are correlated.

Figure 13 depicts details of atmospheric $O_3$ covariances as obtained from model calculations and measurements. The left panels shows how the atmospheric $1\sigma$ variability depends on altitude. Atmospheric covariances can be calculated from atmospheric model outputs or from atmospheric measurements. The obtained covariances strongly depend on the temporal and spatial resolution and the uncertainties of the model or the measurements. The black line shows the variability obtained from monthly mean model data (WACCM) for the atmosphere above the Mexico City basin and for the 1980-2020 time period. The red and blue circles depict the $1\sigma$ variability as obtained from $O_3$ in-situ measurements at UNAM (2.3 km a.s.l.) and Altzomoni (4 km a.s.l.), respectively. There are two circles per location connected by a straight line. The circle representing the low $\sigma$ value is for the $1\sigma$ variability obtained from hourly data measured between 11 and 14 LT, i.e. the local time when FTIR measurements are typically performed. The circle representing the high $\sigma$ value is obtained by taking all hourly measurements between 0 and 23 LT into account (at Altzomoni low and high $\sigma$ values are almost identical). We observe that the measured variability is much larger than the variability as obtained from the model. This high atmospheric variability as seen in the measurements is confirmed for other altitudes by ECC (Electro-Chemical-Cell) radiosondes data. The dashed red line shows the $1\sigma$ variability obtained from measurements in Tenerife, which is similar to Mexico City a subtropical site that offers a very high number of high quality $O_3$ profile in-situ data (Schneider et al., 2005).

The right panel of Figure 13 shows the correlation matrix $\mathbf{\Gamma}$ as obtained from the monthly mean WACCM data. The correlation lengths are longer than those obtained from the Tenerife radiosonde data (see Fig. 7 in Schneider et al., 2005), indicating to limitations in the model's vertical resolution.

## B2  Not measurable covariances

For calculating the covariances that can not be measured by the remote sensing system ($\mathbf{S_{nmc}}$ for "not measurable covariances") we use the formula as suggested by Rodgers (2000) for the "smoothing error":

$$\mathbf{S_{nmc}} = (\mathbf{A} - \mathbf{I})\mathbf{S_{cov}}(\mathbf{A} - \mathbf{I})^T. \tag{B1}$$

We perform calculations for the UNAM, the Altzomoni and the combined product, i.e. for the kernels $\mathbf{A}_{\mathrm{UNAM}}$, $\mathbf{A}_{\mathrm{ALTZ}}$ and $\mathbf{A}_{\mathrm{comb}}$. Furthermore, we use two different atmospheric covariance matrices. The first one is $\mathbf{S_{cov,WACCM}}$ and it is obtained from the monthly mean WACCM data. The second one is $\mathbf{S_{cov,meas.}}$ and it is based on the WACCM data but also considers the higher variations and lower correlation lengths as observed in the measurement data. Compared to $\mathbf{S_{cov,WACCM}}$, the $1\sigma$ variability of $\mathbf{S_{cov,meas.}}$ is increased by a factor of three (except for the boundary layer, where we apply a factor of five) and the correlation length is set to 5 km in the boundary layer (as indicated by Velasco et al., 2008), to 1.5 km up to 35 km and to 3 km above 35 km (in agreement to Schneider et al., 2005).

Figure 14 shows the results. The bright red area indicates the values that are outside the $1\sigma$ variability given by the square root value of the diagonal of $\mathbf{S_{cov}}$. The different lines show the square root values of the diagonal elements of $\mathbf{S_{nmc}}$. The black, grey and red lines show the result for the UNAM product, the Altzomoni product and the combined product, respectively.

For the left panel the calculations are made with $\mathbf{S_{cov}} = \mathbf{S_{cov,WACCM}}$, i.e. it can be used as an estimation of the scatter expected when comparing any individual monthly mean WACCM data from the 1980-2020 time period with a corresponding

remote sensing measurement. However, it should be noted that this calculation does only consider the scatter due to the smoothing characteristics of the remote sensing data. It does not consider the uncertainty in the monthly mean value obtained from irregularly sampled remote sensing data. Above 5 km altitude the uncertainty is smallest in the Altzomoni data (grey line) reflecting their better vertical resolution (compare kernels in Figs. 3 and 5). Below 5 km the uncertainty in the UNAM data

(black line) is smaller because of being sensitive down to 2.3 km, whereas the Altzomoni data have no sensitivity below 4 km. The data of the combined product (red line) have a very low uncertainty below 4 km, but provide no information about the atmospheric state above 5 km (red line enters or comes very close to the bright red area). The high sensitivity of the combined product for $O_3$ variations below 4 km and the lack of sensitivity above 5 km reflects the sharp peak of the 2.3 km row kernel of $\mathbf{A_{comb}}$ (see Fig. 10).

The right panels shows the results for calculations with $\mathbf{S_{cov}} = \mathbf{S_{cov,meas.}}$. It can be used as an estimation of the scatter expected when comparing the remote sensing data to a coinciding individual hourly mean measurement, that has the vertical representativeness as prescribed by the verticla grid of the remote sensing retrieval (the measurement have a vertical resolution of a few meters, but $\mathbf{S_{cov,meas.}}$ only represents the limited vertical grid points of the retrieval). The absolute values are very different to those as depicted in the left panel, but the relative differences between the uncertainty of the UNAM, Altzomoni

and combined products as well as their relative distance to the bright red area (the area marking the values outside the assumed variability) are very similar.

The calculations of the $\mathbf{S_{nmc}}$ covariances reflect the shape of the different averaging kernels and can be resumed as follows: Altzomoni data are better (sharper kernels) for the atmosphere above 5 km than UNAM data (broader kernels), but only UNAM data provide information about the atmosphere below 4-5 km (measurements in Altzomoni are not sensitive to the atmosphere

below 4 km). The combined product is clearly the best choice when the interest lies in the atmosphere below 4 km (sharply peaked 2.3 km kernel), but does not provide information for altitudes above 4-5 km.

## B3   Comparability between different remote sensing data

In Sect. 3.2 we compare the FTIR data obtained from the UNAM and the Altzomoni measurements. We account for the different Altzomoni and UNAM averaging kernels (compare Figs. 3 and 5) by smoothing the Altzomoni data according to Eq. (7). Then

use the metric according to Eq. (8) for identifying situations where the two coinciding remote sensing data cannot be compared due to the significantly different smoothing characteristics.

Since the difference in the kernels is relatively small (typical DOFS values are 3.2 and 4.1 for UNAM and Altzomoni, respectively), one might argue that the data should be compared directly, i.e. without smoothing the Altzomoni data. In order to investigate this issue we calculate an additional metric according to:

$\mathbf{S'_{cmp}} = (\mathbf{A}_{\mathrm{UNAM}} - \mathbf{A}_{\mathrm{ALTZ}})\mathbf{S_{cov}}(\mathbf{A}_{\mathrm{UNAM}} - \mathbf{A}_{\mathrm{ALTZ}})^T,$    (B2)

thereby documenting the comparability when the Altzomoni data are not smoothed.

Figure 15 correlates the metrics obtained from $\mathbf{S'_{cmp}}$ (according to Eq. B2) with the metrics obtained from $\mathbf{S_{cmp}}$ (according to Eq. 8) for all individual coincidences and data that are compared: the total column amounts as well the volume mixing ratios

at 17 km and 32 km a.s.l.. For the volume mixing ratios the comparability values are given by the square root of the diagonals of $\mathbf{S_{cmp}}$ and $\mathbf{S'_{cmp}}$, and for the total column amounts it is the square root of the total column covariances obtained from $\mathbf{S_{cmp}}$ and $\mathbf{S'_{cmp}}$ by applying a total column operator (Rodgers, 2000). The figure reveals that the smoothing of Altzomoni data with the UNAM kernels improves the comparability. There are only very few situations for comparisons of volume mixing ratios at

5    17 km (blue dots), where a direct comparison would be the better choice.

For the here discussed results we use $\mathbf{S_{cov}} = \mathbf{S_{cov,meas.}}$. The calculations according to Eqs. (8) and (B2), are not independent on $\mathbf{S_{cov}}$. They strongly depend on $\mathbf{\Sigma}$ ($\mathbf{S_{cov}} = \mathbf{\Sigma \Gamma \Sigma}^{T}$, see beginning of this Appendix). However, the results do only depend on the correlation length as given by $\mathbf{\Gamma}$ if this correlation length comes close to the width of the smoothing functions given by the averaging kernels. In this respect $\mathbf{S_{cmp}}$ or $\mathbf{S'_{cmp}}$ do not depend on the vertical resolution of the model or measure-

10    ments used for calculating $\mathbf{S_{cov}}$ neither on the vertical grid of the remote sensing retrieval as long as this resolution or grid is finer than the resolution as given by the averaging kernels. This is in contrast to $\mathbf{S_{nmc}}$ which strongly depends on the vertical resolution of the data used for obtaining $\mathbf{S_{cov}}$ as well as on the vertical grid of the remote sensing retrieval.

*Author contributions.* E. F. Plaza-Medina, W. Stremme, A. Bezanilla and M. Grutter operated the Altzomoni and UNAM experiments. M. Grutter coordinated the measurement program at both stations. F. Hase, T. Blumenstock and M. Schneider helped in characterising and operating the Altzomoni FTIR instrument. M. Schneider developed the combined UNAM/Altzomoni remote sensing product. E. F. Plaza-Medina prepared this manuscript with important contributions from all the co-authors.

5    *Acknowledgements.* E. F. Plaza-Medina received funding as a Postdoc at CCA/UNAM (becas mujeres 2012 awarded by ICyTDF). M. Schneider was funded by the European Research Council under the European Community's Seventh Framework Programme (FP7/2007-2013) / ERC Grant agreement number 256961. Financial support from CONACYT (239618, 249374, 275239), DGAPA-UNAM (IN109914, IN112216, IN107417) and BMBF (project 01DN12064) is acknowleded for the maintenance and improvements of the FTIR measurements at Altzomoni and UNAM. The RUOA university network of atmospheric observatories (Red Universitaria de Observatorios Atmosféricos,

10   www.ruoa.unam.mx) provided the infrastructure and logistical support for performing the measurements at both sites. Special thanks to D. Flores, A. Rodríguez, H. Soto and O. Torres for their technical assistence and CONANP authorities for hosting the Altzomoni station at the Izta-Popo National Park. The in-situ $O_3$ data for the stations Pedregal, Santa Ursula and Coyoacan were obtained from the databases of Red Automática de Monitoreo Atmosférico (RAMA), operated by the Ministry of Environment of Mexico City. We acknowledge support by Deutsche Forschungsgemeinschaft and Open Access Publishing Fund of the Karlsruhe Institute of Technology.

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

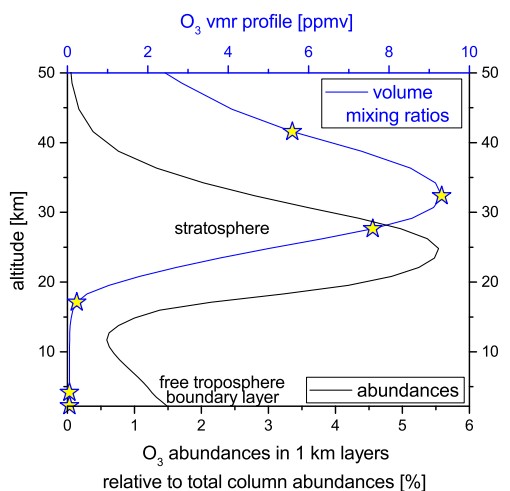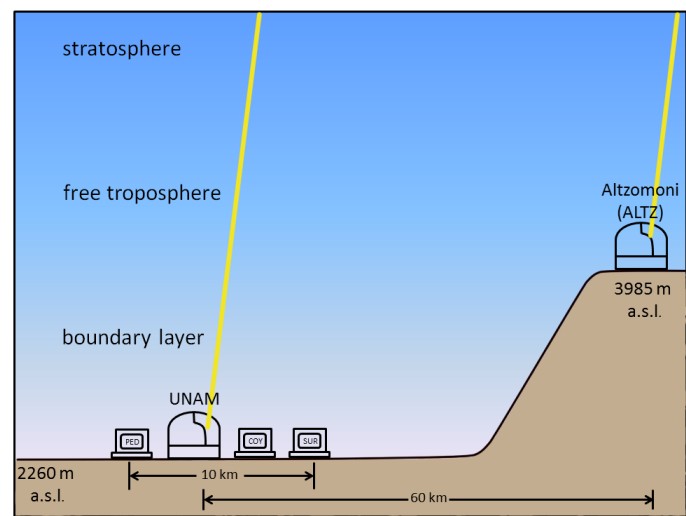

**Figure 1.** Vertical distribution of $O_3$ and location of $O_3$ measuring instrumentation in the surroundings of Mexico City. Left: WACCM version 6 profiles of $O_3$ volume mixing ratios (blue line) and $O_3$ abundances (black line) used as a priori for the FTIR retrieval over Mexico City. The yellow stars indicate altitude regions that are analysed in the following sections. Right: schematics indicating the medium resolution FTIR experiment and the three in-situ instruments (at the stations PED, COY and SUR) in Mexico City at and close to UNAM (at about 2.3 km a.s.l.) and the high resolution NDACC FTIR experiment outside the city in Altzomoni (at about 4 km a.s.l.).

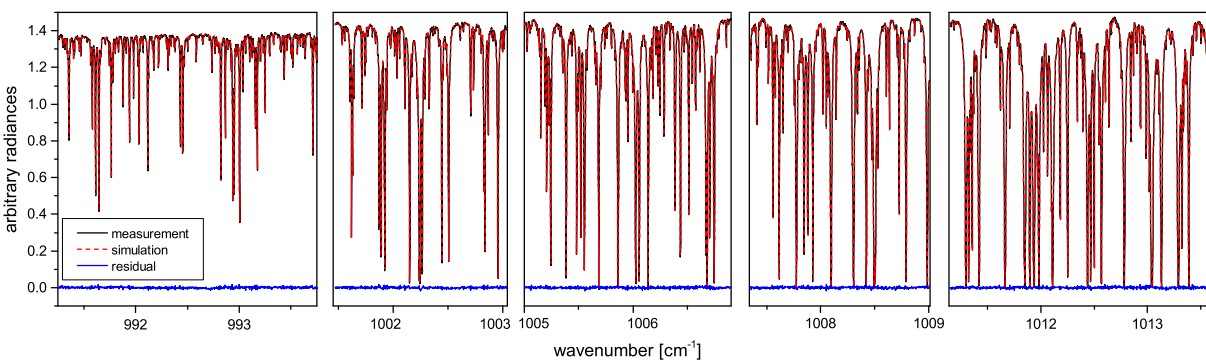

**Figure 2.** The spectral windows used for the retrievals with the NDACC FTIR instrument at Altzomoni. Shown is an example for a typical measurement (22 March 2013, 17:51 UT; solar elevation: 67.9°; $O_3$ slant column: 283 DU). Black line: measurement; red dashed line: simulation; blue line: residual (difference between measurement and simulation).

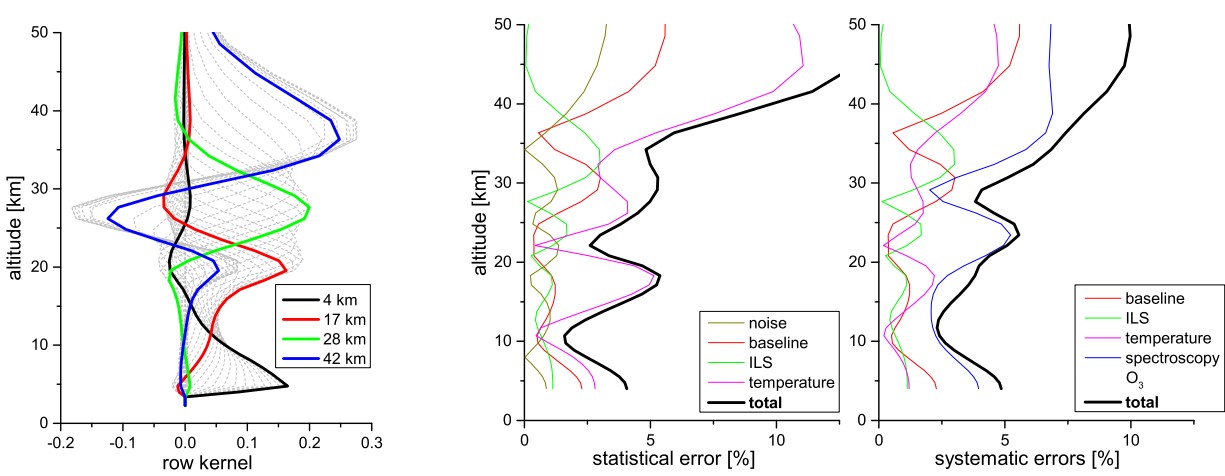

**Figure 3.** Characterisation of the $O_3$ volume mixing ratio profiles obtained from the FTIR measurements at Altzomoni as shown in Fig. 2. Left: row averaging kernels, whereby kernels for a few different altitudes are highlighted by different colours. Right: Estimated statistic and systematic errors, whereby the errors resulting from the different uncertainty sources as listed in Table 1 are represented in different colours. The total error is the root-sum-square of the individual errors and is shown as thick black line.

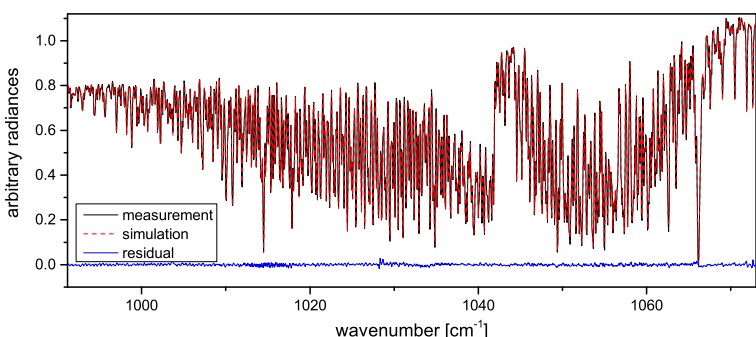

**Figure 4.** Same as Fig. 2 but for the spectral window used for the retrievals with the FTIR instrument at UNAM. Shown is an example for 22 March 2013, 17:57 UT; solar elevation: 68.2°; $O_3$ slant column: 285.4 DU.

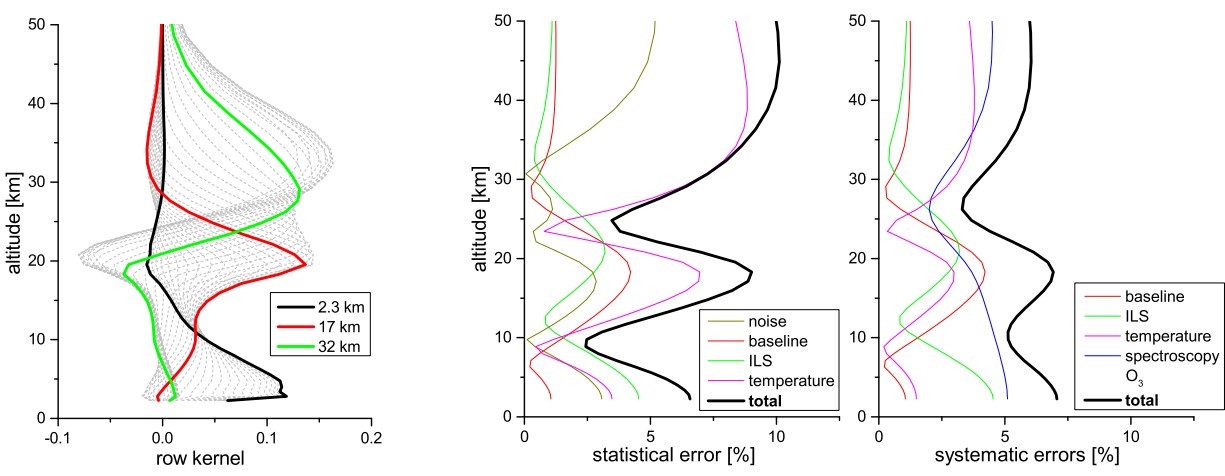

**Figure 5.** Same as Fig. 3, but for the profiles obtained from the medium resolution FTIR measurements at UNAM as shown in Fig. 4.

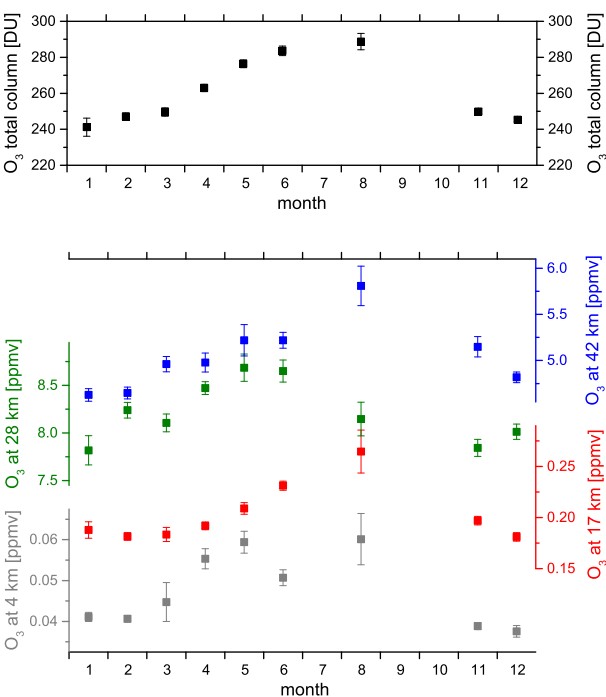

**Figure 6.** Seasonal O₃ cycles as obtained from the Altzomoni FTIR measurements. Shown are means of all the months with measurements for at least three different days for the altitudes that are highlighted in the kernel plot of Fig. 3 (bottom panel) and for the total column amounts (top panel). Please note that for November-February the monthly means are calculated with data from two different years (2012/13 and 2013/14) for the other months the monthly means are calculated from data of 2013 only.

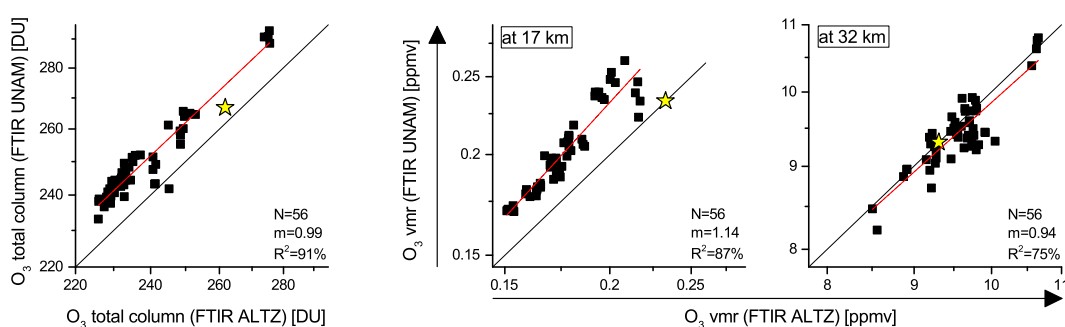

**Figure 7.** Correlation of $O_3$ data products obtained from the Altzomoni and UNAM FTIR measurements. The yellow stars represent the a priori values, the black lines are the 1-to-1 diagonals and the red lines are the fitted linear regression lines. Left: total column abundances. Right: volume mixing ratios at 17 and 32 km a.s.l. (in the UNAM kernel plot of Fig. 5 the respective kernels are highlighted by red and green colour). Please note that for this comparison the Altzomoni data have been smoothed by the UNAM averaging kernels according to Eq. (7).

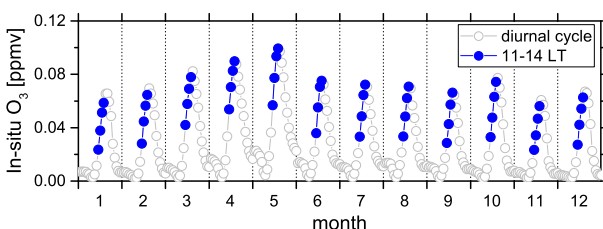

**Figure 8.** Seasonality of the diurnal cycles of $O_3$ volume mixing ratios as measured by the three in-situ monitoring stations at about 2.3 km a.s.l. close to UNAM. The blue dots indicate the time frame when FTIR measurements at UNAM have been typically performed.

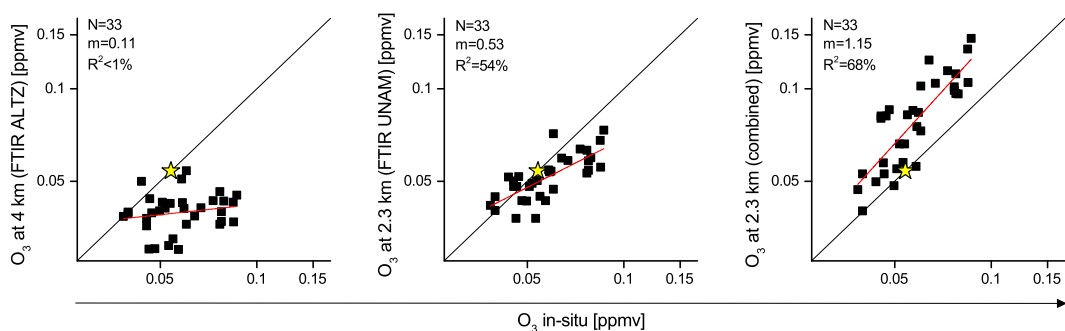

**Figure 9.** Correlation between $O_3$ volume mixing ratios measured by the in-situ instruments for the boundary layer of Mexico City and the ratios obtained by the remote sensing experiments. The yellow stars represent the a priori values, the black lines are the 1-to-1 diagonals and the red lines are the fitted linear regression lines. Left: in-situ $O_3$ versus $O_3$ obtained from the Altzomoni FTIR measurements for 4 km a.s.l.. Middle: in-situ $O_3$ versus $O_3$ obtained from the FTIR UNAM measurements for 2.3 km a.s.l.. Right: in-situ $O_3$ versus the combined FTIR $O_3$ product for 2.3 km a.s.l. (calculation according to Eq. 10).

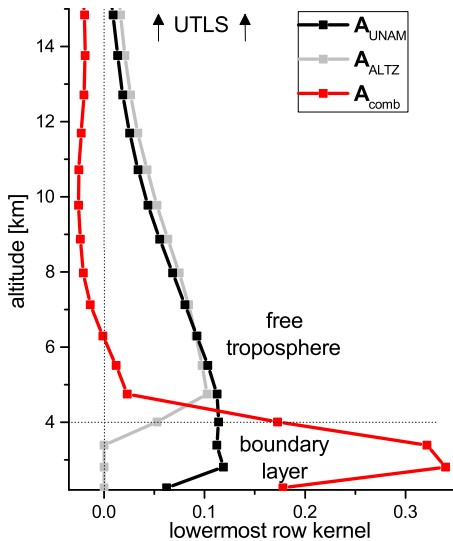

**Figure 10.** O$_3$ row kernels for the lowermost altitude of different FTIR products. Black: 2.3 km row kernel for the UNAM product; Grey: 4 km row kernel for the Altzomoni product; Red: 2.3 km row kernel for the combined product.

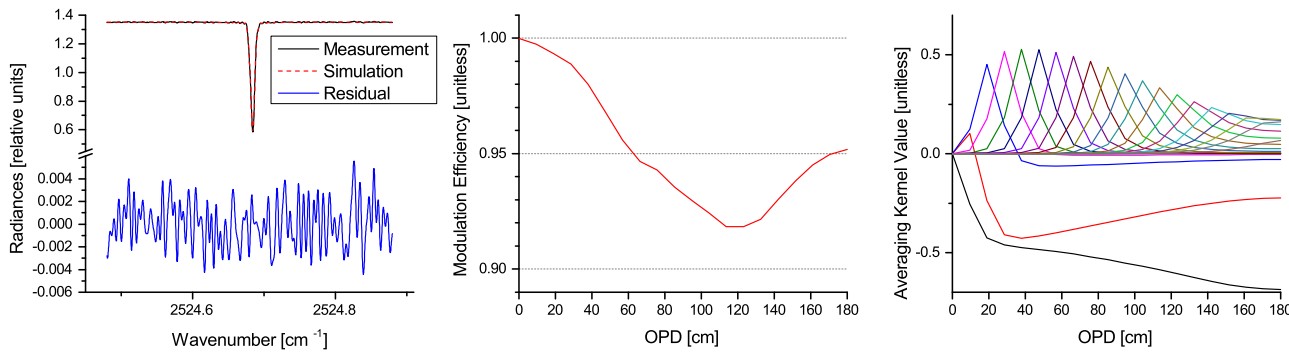

**Figure 11.** Typical ILS analysis for the IFS 120/5 at Altzomoni. Left panel: Example of cell spectrum and analysis; Middle panel: retrieved modulation efficiency; Right panel: averaging kernel for the modulation efficiency.

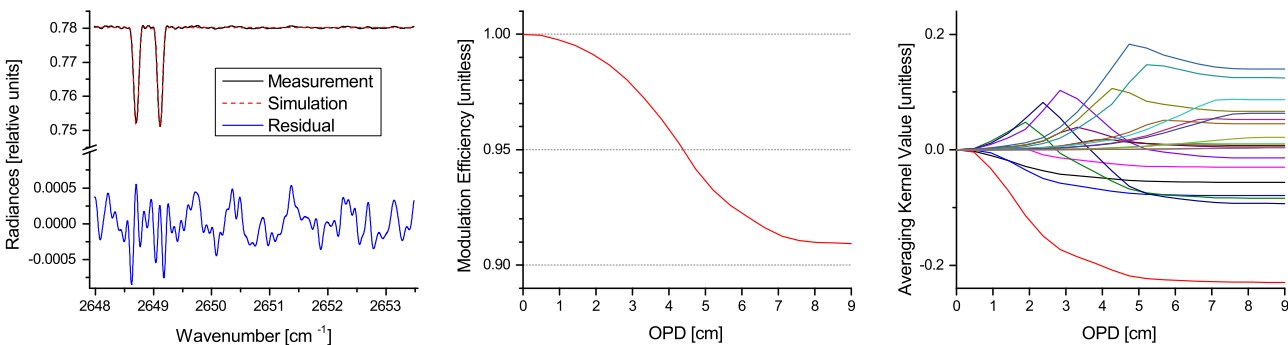

**Figure 12.** Same as Fig. 11, but for the Vertex 80 at UNAM.

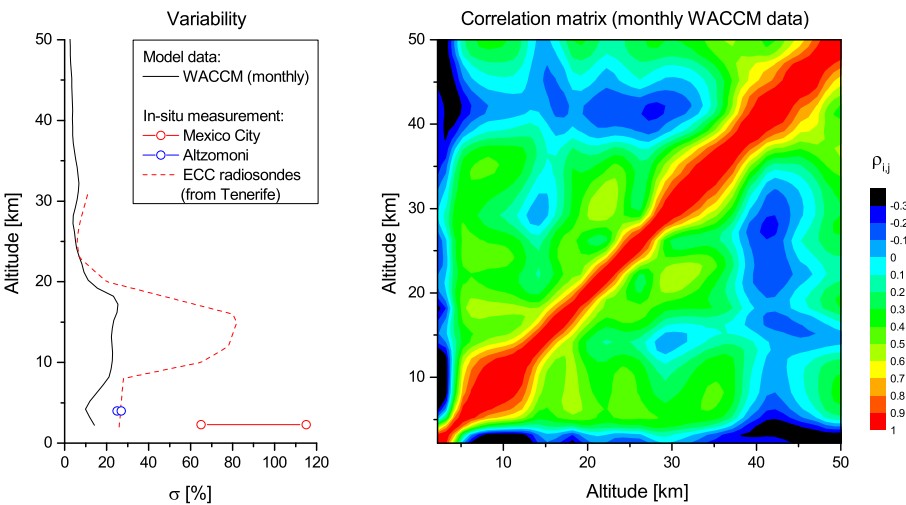

**Figure 13.** Atmospheric $O_3$ covariances ($\mathbf{S_{cov}} = \mathbf{\Sigma}\mathbf{\Gamma}\mathbf{\Sigma}^{T}$) as obtained from models and measurements. Left: diagonal elements of $\mathbf{\Sigma}$ ($1\sigma$ variability), as obtained from WACCM (black line), ECC radiosondes in Tenerife (red dashed line) and surface in-situ measurements in Altzomoni (blue circles) and Mexico City (red circles). Right: correlation matrix $\mathbf{\Gamma}$ as obtained from WACCM.

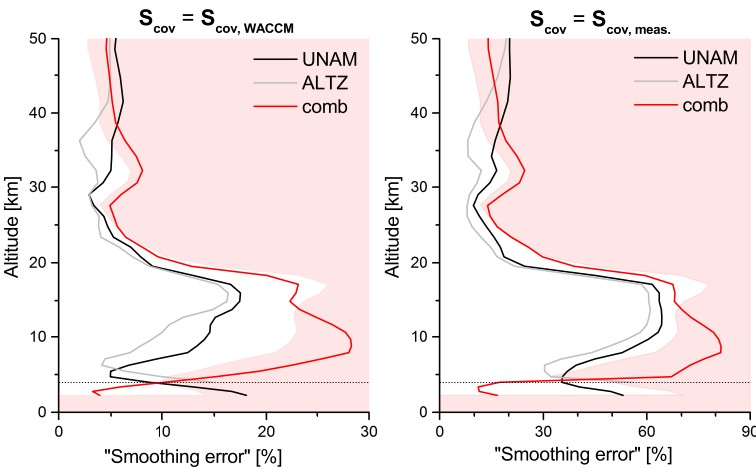

**Figure 14.** Square root values of the diagonal elements of $\mathbf{S_{nmc}}$ as obtained according to Eq. (B1) by using two different $\mathbf{S_{cov}}$. Left for using $\mathbf{S_{cov}} = \mathbf{S_{cov,WACCM}}$, i.e. assuming atmospheric $O_3$ covariances according to the model WACCM. Right for using $\mathbf{S_{cov}} = \mathbf{S_{cov,meas.}}$, i.e. assuming atmospheric $O_3$ covariances in agreement to in-situ measurements. Black line: for UNAM retrieval products; Grey line: for Altzomoni retrieval products; Red line: for the combined product. The bright red area indicates the values that are larger than the assumed atmospheric variation.

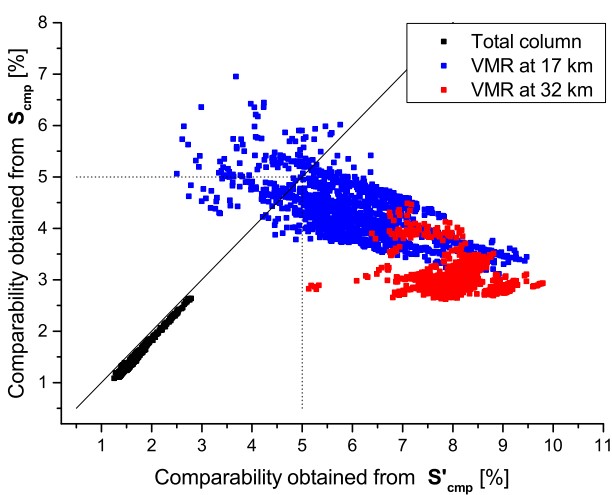

**Figure 15.** Comparability between the UNAM and Altzomoni remote sensing data, calculated according to Eqs. (8) and (B2). Shown the values for three products: the total column amounts (black), the VMRs at 17 km and the VMRs at 32 km.

**Table 1.** Uncertainties used for the error estimation. The second column gives the uncertainty value and the third column the partitioning between statistical and systematic sources.

| Source | Uncertainty | Statistical/Systematic |
|---|---|---|
| Measurement noise | 0.3% (for Altzomoni) | 100/0 |
| | 0.5% (for UNAM) | 100/0 |
| Baseline (channeling, assuming 4 frequencies: 0.005, 0.2, 1.0, and $3.0\,\mathrm{cm}^{-1}$) | 0.2% | 50/50 |
| Baseline (offset) | 1% | 50/50 |
| Instrumental line shape (mod. eff. and pha. err.) | 5% and 0.1 rad | 50/50 |
| Temperature profile | 1.5 K (surface - 12.5 km a.s.l.) | 70/30 |
| | 1.5 K (12.5 - 45 km a.s.l.) | 70/30 |
| | 6 K (above 45 km a.s.l.) | 70/30 |
| Line of sight | 0.2° | 90/10 |
| Solar lines (intensity and $\nu$-scale) | 1% and $10^{-6}$ | 80/20 |
| Spectroscopic parameters of $O_3$ | 2% (line intensity) | 0/100 |
| | 5% (pressure broadening) | 0/100 |
| Interference with water vapour | 100% (atmospheric $H_2O$) | 50/50 |

**Table 2.** Estimated errors for the total column abundances of $O_3$ remote sensing data obtained at Altzomoni and UNAM.

| Error source | Altzomoni Stat. / Sys. | UNAM Stat. / Sys. |
|---|---|---|
| Measurement noise | 0.1% / – | 0.4% / – |
| Baseline | 0.9% / 0.9% | 1.0% / 1.0% |
| Instrumental line shape | 0.4% / 0.4% | 1.0% / 1.0% |
| Temperature | 2.4% / 1.0% | 2.5% / 1.1% |
| Line of sight | 0.1% / <0.1% | 0.1% / <0.1% |
| Solar lines | <0.1% / <0.1% | <0.1% / <0.1% |
| $O_3$ spectroscopy | – / 2.0% | – / 2.0% |
| Interference with $H_2O$ | <0.1% / <0.1% | <0.1% / <0.1% |
| **Total** | **2.6% / 2.5%** | **2.9% / 2.7%** |

**Table 3.** Estimated errors for the boundary layer $O_3$ remote sensing (retrieval results for 2.3 km altitude). Listed are statistical and systematic errors of the UNAM product and the combined product.

| Error source | UNAM product Stat. / Sys. | Combined product Stat. / Sys. |
|---|---|---|
| Measurement noise | 3.1% / – | 6.8% / – |
| Baseline | 1.0% / 1.0% | 14.4% / 14.4% |
| Instrumental line shape | 4.5% / 4.5% | 13.1% / 13.1% |
| Temperature | 3.5% / 1.5% | 5.5% / 2.4% |
| Line of sight | 0.1% / <0.1% | 0.4% / <0.1% |
| Solar lines | <0.1% / <0.1% | <0.1% / <0.1% |
| $O_3$ spectroscopy | – / 5.1% | – / 15.9% |
| Interference with $H_2O$ | <0.1% / <0.1% | <0.1% / <0.1% |
| **Total** | **6.6% / 7.1%** | **21.4% / 25.2%** |