# Peer review of "Ground-based remote sensing of O3 by high and medium resolution FTIR spectrometers over the Mexico City basin"

_Atmospheric Measurement Techniques, 2017_

## Referee Comment (RC1) · Anonymous Referee #2 · 3 Apr 2017

**General comments**

This paper presents for the first time FTIR ozone measurements over Latin America, at 2 close but different sites (Altzomoni, a mountain observatory; and Mexico city, a polluted site). Note that ozone FTIR data are available in the NDACC database at Paramaribo (Suriname, 5°48'N, 55°12'W), but to my knowledge these results have not been published. Furthermore, the authors combine the measurements at these 2 sites to obtain a new product: an O3 boundary layer. This could provide interesting new information and would be promising for future investigation of the Mexico city pollution. Therefore, I recommend the publication of this paper in AMT. However, I have some few questions to be clarified, and some specific comments that need to be taken into

account before publication. The 2 main points being the fact that the authors forgot about the smoothing error in their uncertainty budget, and that their "new product" is not convincing to me at present. So it should be at the very least clarified, or omitted in the new manuscript.

**Specific comments/questions**

1) Section 2.2 (Retrieval method) and 2.3 (Error analysis)

a) Are the baseline (channeling and offset), the ILS, the temperature, the line of sight, and the solar lines fixed model parameters or do you retrieve them ? In the case of UNAM, the Linefit results (Hase, 2012) are given as input in the forward model or Linefit is just used to check the alignment (you specify only for Altzomoni) ?

b) You wrote that all interfering species are from WACCM. Does that mean that this is also the case for $H_2O$ ? Maybe the 100% uncertainty taken for $H_2O$ interference could be reduced by using e.g. either preliminary retrieval of $H_2O$ or 6-hourly $H_2O$ profiles from NCEP. However, you obtain very small errors due to $H_2O$ in Table 2, so this seems not so important at your sites anyway.

c) Spectroscopic errors: p.6, l.4-5, you say that your spectroscopic errors are coming from the HITRAN line list and in the discussion p. 16, l.16-17, you say that the uncertainty on pressure broadening parameters could be as large as 20% (instead of 5%). Could you explain ? Does this mean that the 5% should not be trusted ?

d) Table 1: I don't understand how the numbers in the column "Statistical/Systematic" are obtained, and how they are applied in the error analysis. Can you explain in the paper ?

e) My main concern about this paper is that one of the dominating random error source is missing in your budget (Tables 2  3; Figs. 3 and 5): the smoothing error. You need to replace your Eq. 2 by, e.g., Eq. 1 of Schneider et al. ACP 2008 ("Quality assessment of O3 profiles..."). And calculate the smoothing error, which is clearly dominant on

ozone profiles and partial columns.

2) Section 3: Free tropospheric and stratospheric O3

a) Seasonal cycle at UNAM: you could add the seasonal cycle of the total columns at UNAM in Fig.6 (top panel), for comparison. Same for the different altitudes, if you define common altitudes where both instruments have information. (as you did for Fig.7).

b) Fig. 6 (lower panel): I would prefer 4 different subplots instead of all curves in the same plot. You could also add UNAM seasonal cycle (see comment a) ).

c) Seasonal cycle discussion: since you have a model available (waccm), I wonder if the model reproduces well your seasonal cycles. It could be interesting to check this, and maybe add the model seasonal cycle on the plots.

d) For me, since FTIR has low vertical resolution, it would be better to describe your data in partial columns (PC) where you have about one DOFS, rather than at a given altitude point. If you take into account the smoothing error, you will find a larger error on a single altitude point (the dominant random error on the profiles is the smoothing error), than on a PC. This division in PC based on DOFS is commonly used in FTIR studies (e.g. García et al., 2012; Vigouroux et al., 2015; Duchatelet et al., 2010;...). I would use 4 PC in Fig. 6 (lower panel) instead of altitude points, and also in Fig.7. Your agreement between Altzomoni and UNAM PC would be probably better using common PC rather than with single altitude points (Sect. 3.2).

e) Scmp formula (Eq.8): I think you should give the reference for this formula (Rodgers and Connor, 2003). In Rodgers and Connor (2003), Scmp is not only your Eq. 8, you also have the terms $S_{UNAM} + A_{UNAM} S_{ALTZ} A_{UNAM}$ in addition (Eq. 30 of Rodgers and Connor, 2003). Since you do not use Eq. 8 in this Sect. 3.2, I am not sure that you are indeed talking about the complete error covariance matrix of the comparison between the 2 instruments ("error on the differences", are you ? If yes, then you should

use the complete Eq. 30 of Rodgers and Connor (2003).

f) In principle, you should use $S_{cmp}$ (complete formula) to conclude if the 2 instruments are in agreement: standard deviation of the differences (1.2%; 5.2%, and 2.1% for your 3 comparisons) < "random error on the differences". The bias of 8.7% at 17 km is probably not explained by the systematic uncertainty budget on the differences (same spectroscopy, same temperature profile), but this bias might improve if you use PC instead of a point at 17 km.

g) p.11, l. 15: Scov= 100% O3 variability: this is a large assumed variability, especially in the stratosphere. I think you are overestimating then your error on comparison. Why not taking Scov from the model WACCM ? (or any other climatology, e.g. from satellite?)

h) p.11, l. 16-17: I don't understand how you can "require that the square root of... is smaller than 10%". If you fix Scov (100% O3 variability + 5 km correlation), and you have $A_{UNAM}$ and $A_{ALTZ}$ fixed from your retrievals, how can you control $S_{cmp}$ ? Sorry, I am missing what you mean here.

i) p.11, l. 26-27: you said that the bias of 2% is due to the different altitudes between the 2 instruments. But if I get right how you construct $x^*_{ALTZ}$, you are not using zero values from 2.3 and 4km, but the values of xa. Then you should not have some bias (at least not so large). But maybe you calculate from $x^*_{ALTZ}$ a total column that starts only at 4 km ? If yes, please specify in the text.

3) Section 4: Boundary layer

a) Sect. 4.2: Again here, you could reduce the error on the FTIR products by using PC instead of a single value at 2.3 or 4 km (as in, e.g., Sepúlveda et al., 2014) ? These values are correlated to the whole tropospheric column anyway. I would be curious to see Fig. 9 with PC instead of single altitude values.

b) Sect. 4.3: p.13, l.24-30: Note that if you use Eq.30 of Rodgers and Connor (2003),

you directly have the combined error on the differences, i.e., your 4-5% value.

c) I don't understand why having a boundary layer product coming from the total columns differences with an error of about 4-5%, would be worse than the combined product that you propose for which you reach 21% ! Therefore, I am really surprised that the correlation with in situ data using your combined product is better than using the total column differences.

d) I am not convinced by the construction of the combined product. So I strongly suggest that you document better what you are doing. Is there any reference that could be added to your calculation ? At present these iterations between Eq. 9 and 11 (why only 2 ?) seem arbitrary. I have the feeling that the DOFS are increased "arbitrarily" in the boundary layer (p.15, l. 29 – p.26, l.1; and p.26, l.9-10). There is one instrument at 2.3 km with "little sensitivity up to 4 km (from 0.05 to 0.12 Fig.10), and one instrument that is not measuring below 4 km, so to me what you are doing by using the information above 4 km from the 2 instruments, removing it (Acom come to about zero above 4 km; Fig 10), and transferring the information to the 2.3-4km is equivalent to use the differences between the 2 instruments. And if you want to do so, the more precise way should be to use the total columns since the errors of the instruments are the lowest for total columns, because the smoothing error is much reduced for total columns. It might be that I am missing something, but then more information and references are needed in the paper to explain why this combined product is not only "artificial", and that real information is appearing in this boundary layer, with an added value compared to differences of total columns.

**Technical / minor comments:**

- Abstract, l. 6: "three" and "four" should be inverted.

- p.2, l. 21: " of Paris, Viatte et., 2011)" : add the comma.

- p.2, l. 22: Missing "." And "However" instead of "however".

- p.2, l. 26-27: A new FTIR station is measuring now in Brazil (Porto Velho), since July 2016. Although it is not yet an "NDACC" station, I think it's worth to mention it.

- p.5, l. 22: "listed in in Table 1": remove one "in".

- p.6, l. 8: "Observations at Altzomoni" (not Atzomoni)

- p.13, l.30: "expected", not expect.
* * *

---

## Author Comment (AC1) · 7 Apr 2017

Dear referee,

Many thanks for commenting on our manuscript. Please see our reply in the attached pdf-file.

Best regards.

Please also note the supplement to this comment:
http://www.atmos-meas-tech-discuss.net/amt-2017-7/amt-2017-7-AC1-supplement.pdf

---

## Referee Comment (RC2) · O. Garciar (Referee) · 14 Apr 2017

General Comments

This paper presents for the first time the time series of the vertical profiles and total column amounts of one of the most important gases on the atmospheric chemistry, ozone (O3), for Latin America. The strategic location of the ground-based stations used and the methodology proposed, using high-resolution and medium-resolution Fourier Transform infrared (FTIR) solar absorption spectra, provide high confidence to the results obtained. Especially interesting is the presentation of an improved tropospheric ozone product from the combination of two remote sensing FTIRs data, which allows for a better monitoring of tropospheric ozone concentrations. The paper is well-written

and concise. Thereby, I suggest this paper may be suitable for publication after addressing the specific comments listed below.

Specific Comments

Section 2: Ground-based FTIR remote sensing

a) The authors mention that the microwindows used to retrieve the O3 concentrations are the same windows as presented in Schneider and Hase (2008), but this is not strictly true. Schneider and Hase (2008) suggest a broad microwindow between 1000 and 1005 cm-1, which has been split into two microwindows in the current work. I guess that the authors did this to avoid the interference of water vapour (H2O). If so, please clarify in the text. Also, the authors include the 1005-1006cm−1 microwindow, not present in Schneider and Hase (2008). So, I think it would be more appropriate to say that the selection of the O3 microwindows is based on Schneider and Hase (2008) and clarify why the authors modify the original microwindows.

b) The error budgets clearly show that the atmospheric temperature profile is an important error source for Altzomoni and UNAM FTIRs as well as for the combined product. Simultaneous temperature retrieval with O3 profile has widely demonstrated that improves theoretically and experimentally the quality of the FTIR O3 products (e.g. Schneider and Hase, 2008, Schneider et al., 2008, García et al., 2012). Have the authors considered performing this temperature retrieval? Why not?

c) The authors have assumed the same error in the ILS for both FTIRs (5% for the modulation efficiency and 0.1 rad for the phase error). But, as for the measurement noise, I guess that the ILS's error of the medium-resolution FTIR should be a bit higher than the high-resolution one. Also, the ILS's errors could explain in part the large systematic/statistical errors observed in the combined product. Why the authors use the same value? Why the authors do not consider the ILS as a possible explanation for the large errors found in the combined product? Could the authors show a plot with the time series of the ILS for each station? What is the frequency of the cell measurements

at each station?

d) How the layers that are detected by each FTIR are defined? Are the consecutive levels added until a DOF of one? There is a high signal in the UNAM averaging kernels at about 35 km, what is the reason?

Section 3: Free tropospheric and stratospheric O3

a) I am wondering if smoothing the Atlzamoni averaging kernels with the UNAM ones makes sense when the difference of the total DOFs is only of one. Would it be more appropriate to estimate the altitudes that are well comparable analyzing the square root of the diagonal of Scmp as presented in Wiegele et al. (2014) and then compare the original volume mixing ratios? How are the correlations shown in Fig. 7 for the not-smoothing data?

b) The authors use the works of Thompson et al. (2008) and Emmons et al. (2010) to explain in part the observed O3 annual cycle of the free troposphere. These works point out the importance of the transport of the urban emissions on the free tropospheric O3 levels. But, according to the authors in page 7, line 32, "the Altzomoni solar absorption spectra are only very weakly affected by the polluted boundary layer". So, I find both statements contradictory, but maybe I am missing something. Have the authors considered that the maxima in spring-summer could indicate the importance of photochemical production of tropospheric ozone?

c) The O3 annual cycles at Altzomoni are compared with other NDACC stations such as Izaña or Jungfraujoch. But, there is another NDACC site at equal latitude and about the same altitude, Mauna Loa. Have the authors compared the Altzomoni and Mauna Loa FTIR O3 products? This could be a very nice cross-validation of the Altzomoni O3 data, especially for the stratospheric values.

d) The authors estimate the Scmp to filter out the data using a value of 10% for the whole range of altitudes. But, this threefold could be altitude-dependent as suggested

in Fig. 7, where a worse correlation is observed for the upper level. Plots showing the Scomp vs correlation and the profiles of the square root of the diagonal of Scmp could help to analyze better the comparability of the two remote sensing instruments. See comment a) related to use the Scmp only for filtering the data and not to look for the altitudes for direct comparison.

Section 4: Boundary layer

a)The theoretical error budget for the combined product is not in agreement with the experimental comparison (higher statistical errors and lower systematic ones). Also, in the abstract the authors mention that the combined products offer theoretically and experimentally better results. This is true for the sensitivity, but not for the error estimation shown, which could cause some confusion to the readers. The authors provide some causes to account for these discrepancies, but they have considered re-doing the error estimations with more realistic values. For example, is it enough to assume 5% and 0.1 rad for the ILS errors when the UNAM ILS is assumed as ideal?

b) As the authors stated, the combined product is very promising. I am interested in other possible applications of this product. Could it be used the other way round, ie, to reduce the tropospheric signal in the Atlzomoni ULTS data? Or to improve the tropospheric sensitivity of space-based sensors such as IASI?

Technical Comments Page 5, line 12. Please include reference for the WACCM model-version 6. Page 7, line 14. Replace "in temperature" by "temperature". Page 10, line 8. Replace "subtropical" by "sub-tropical". Page 12, line 5. Replace "in situ" by "in-situ". Page 16, line 11. Replace "Eq. 10" by "Eq.(10)".

---

## Author Comment (AC2) · 18 May 2017

**General Comments**

This paper presents for the first time the time series of the vertical profiles and total column amounts of one of the most important gases on the atmospheric chemistry, ozone (O3), for Latin America. The strategic location of the ground-based stations used and the methodology proposed, using high-resolution and medium-resolution Fourier Transform infrared (FTIR) solar absorption spectra, provide high confidence to the results obtained. Especially interesting is the presentation of an improved tropospheric ozone product from the combination of two remote sensing FTIRs data, which allows for a better monitoring of tropospheric ozone concentrations. The paper is well-written and concise. Thereby, I suggest this paper may be suitable for publication after addressing the specific comments listed below.

**Specific Comments**

**Section 2: Ground-based FTIR remote sensing**

a) The authors mention that the microwindows used to retrieve the O3 concentrations are the same windows as presented in Schneider and Hase (2008), but this is not strictly true. Schneider and Hase (2008) suggest a broad microwindow between 1000 and 1005 cm-1, which has been split into two microwindows in the current work. I guess that the authors did this to avoid the interference of water vapour (H2O). If so, please clarify in the text. Also, the authors include the 1005-1006cm−1 microwindow, not present in Schneider and Hase (2008). So, I think it would be more appropriate to say that the selection of the O3 microwindows is based on Schneider and Hase (2008) and clarify why the authors modify the original microwindows.

Yes, the referee is right. We will correct and clarify this in the revised version. Thanks!

b) The error budgets clearly show that the atmospheric temperature profile is an important error source for Altzomoni and UNAM FTIRs as well as for the combined product. Simultaneous temperature retrieval with O3 profile has widely demonstrated that improves theoretically and experimentally the quality of the FTIR O3 products (e.g. Schneider and Hase, 2008, Schneider et al., 2008, García et al., 2012). Have the authors considered performing this temperature retrieval? Why not?

Simultaneous temperature retrievals can improve the quality of the retrieved O3 data using high resolution spectra and spectral windows in the 1000cm-1 region. This has been shown almost 10 years ago (Schneider and Hase, 2008) and has been confirmed by other studies (e.g., García et al., 2012). There are two reasons why for this study we decided not performing simultaneous temperature retrievals. First, for the retrieval setup that uses the medium resolution spectra, so far no simultaneous temperature retrieval study has been performed. So we would need to do such study in this paper (which would be an extra paper and it is out of the scope of the here presented work). Second, simultaneous temperature retrievals mean an advanced retrieval methodology (for instance, the simultaneous temperature fit interferes with ILS uncertainties) and for a first study with the Altzomoni data, we prefer using a more common and less sophisticated retrieval methodology. Actually, despite the clearly demonstrated benefit of the simultaneous temperature retrievals, it has so far only been used in a rather limited number of studies, mainly those made with the FTIR instrument at the Izana Observatory. The

advanced retrieval methodology has for instance not been used for most of the retrievals, whose data are presented in the trend studies as shown in Vigouroux et al. (2015).

c) The authors have assumed the same error in the ILS for both FTIRs (5% for the modulation efficiency and 0.1 rad for the phase error). But, as for the measurement noise, I guess that the ILS's error of the medium-resolution FTIR should be a bit higher than the high-resolution one. Also, the ILS's errors could explain in part the large systematic/statistical errors observed in the combined product. Why the authors use the same value? Why the authors do not consider the ILS as a possible explanation for the large errors found in the combined product? Could the authors show a plot with the time series of the ILS for each station? What is the frequency of the cell measurements at each station?

The medium resolution spectra are measured with a maximal optical path difference (OPDmax) of 9cm, whereas the high resolution spectra are measured with an OPDmax of 180cm. So the requirement of the modulation efficiency being stable within 5% along 180cm is significantly stricter than the requirement of being stable within 5% along 9cm, only. So we actually assume that the high resolution FTIR is much more stable than the medium resolution FTIR. The uncertainty values of 5% are estimated according to different cell measurements made at Altzomoni and UNAM.

We are aware that ILS monitoring is very important, especially for retrieving stratospheric trace gases with high precision. In this context the work done at Izana (see for instance the ILS time series as shown in García et al., 2012) is a nice reference and can serve as guideline for other stations. We strongly support the idea that publication of FTIR data should be accompanied by a documentation of the ILS. In the Appendix we will show ILS retrievals and respective averaging kernels and hope that this will become a standard for future publications of data from new stations (the following Fig. 1 will be added in an Appendix of the revised paper).

[Figure]

*Fig. 1: Cell measurements (examples of spectral windows) and corresponding ILS retrieval results (modulation efficiency) for the medium resolution instrument at UNAM (top) and the high resolution instrument at Altzomoni (bottom).*

Referee #2 also commented on the ILS at UNAM. For simplicity we have used nominal ILS during the retrieval process, however, the referees are right and the loss of modulation efficiency is also not negligible for the medium resolution instrument. So we will redo the UNAM medium resolution retrievals using the measured ILS.

d) How the layers that are detected by each FTIR are defined? Are the consecutive levels added until a DOF of one? There is a high signal in the UNAM averaging kernels at about 35 km, what is the reason?

We are not sure if we correctly understand this comment. We look on the full averaging kernels matrix not only on the diagonal elements (the diagonal elements of the averaging kernel matrix are used for calculating the DOFS).
Does the referee ask about our criteria for choosing the altitudes 4, 17, 28, and 42 km (for Altzomoni) and 2.3, 17, and 32 km (for UNAM)? We simply chose these altitudes by looking on the row averaging kernels (Figs. 3 and 5), because the row kernels reveal that the data retrieved for these altitudes reflect rather different altitudes (there is no significant overlap of the respective kernels).
The averaging kernels depend on the quality of the measured spectra (spectral resolution and signal-to-noise ratio) and the kind and strength of the constraint. They document how the remote sensing system interprets real atmospheric variability.

Section 3: Free tropospheric and stratospheric O3

a) I am wondering if smoothing the Atlzamoni averaging kernels with the UNAM ones makes sense when the difference of the total DOFs is only of one. Would it be more appropriate to estimate the altitudes that are well comparable analyzing the square root of the diagonal of Scmp as presented in Wiegele et al. (2014) and then compare the original volume mixing ratios? How are the correlations shown in Fig. 7 for the not-smoothing data?

This is a good point and we were also concerned about that. So we compared the raw row kernels and the row kernels after smoothing and found that smoothing is an advantage by making the kernels better comparable (see also the following Figure).

[Figure]

*Fig. 2: Row averaging kernel, examples for 4, 20 and 30km, for UNAM (red), Altzomoni (black), and Altzomoni smoothed by UNAM kernels (green).*

b) The authors use the works of Thompson et al. (2008) and Emmons et al. (2010) to explain in part the observed O3 annual cycle of the free troposphere. These works point out the importance of the transport of the urban emissions on the free tropospheric O3 levels. But, according to the authors in page 7, line 32, "the Altzomoni solar absorption spectra are only very weakly affected by the polluted boundary layer". So, I find both statements contradictory, but maybe I am missing something. Have the authors considered that the maxima in spring-summer could indicate the importance of photochemical production of tropospheric ozone?

Yes, the increase of free tropospheric O3 background in spring is partly connected to O3 pollution transported from the boundary layer of Mexico City (see discussions in Thompson et al., 2008 and Emmons et al. 2010). In the data of our paper we see an increase in the peak values of the surface in-situ data between January and May (see Fig. 8). This is consistent to the increase observed in the Altzomoni retrievals at 4km (see Fig. 6). However, the dominating boundary layer signal has a diurnal time scale. For this reason we observe no significant correlation between the coinciding surface in-situ data and the Altzomoni retrievals at 4km (see left panel in Fig. 9). So the Altzomoni data do not reflect this strong diurnal cycle, especially since the measurements are made mainly before 14:00, when the boundary layer top altitude is below or only slightly above the altitude of Altzomoni. Nevertheless the 4km O3 background levels likely depend on the outflow of O3 pollution from the boundary layer. We will improve the respective discussion in the text.

c) The O3 annual cycles at Altzomoni are compared with other NDACC stations such as Izaña or Jungfraujoch. But, there is another NDACC site at equal latitude and about the same altitude, Mauna Loa. Have the authors compared the Altzomoni and Mauna Loa FTIR O3 products? This could be a very nice cross-validation of the Altzomoni O3 data, especially for the stratospheric values.

In Sect. 3.1 we mention in one sentence that there are similarities in the annual cycles between Altzomoni (on the one hand) and Izana and Jungfraujoch (on the other hand). The idea is to give references to studies that make scientific analyses of similar FTIR O3 data. We mention these two stations because there are a lot of publications available. We can in addition search for publications with Mauna Loa FTIR O3 data and cite it in the same sentence.

A comparison of time series obtained at different stations or a discussion of model measurement differences is certainly interesting. However, this should be in our opinion addressed in an additional ACP paper (instead of AMT). Here we focus on the retrieval technique and data validation.

d) The authors estimate the Scmp to filter out the data using a value of 10% for the whole range of altitudes. But, this threefold could be altitude-dependent as suggested in Fig. 7, where a worse correlation is observed for the upper level. Plots showing the Scomp vs correlation and the profiles of the square root of the diagonal of Scmp could help to analyze better the comparability of the two remote sensing instruments. See comment a) related to use the Scmp only for filtering the data and not to look for the altitudes for direct comparison.

We set up a Scov having $100^2\%^2$ in the diagonal. Then we require that for a meaningful comparison the diagonal elements of Scmp should be smaller than $10^2\%^2$. Thereby we require that as a maximum 10% of the variance between the two data can be explained by different smoothing characteristics. The rest of the scatter can be explained by errors. We apply no extra

filter for errors, because we want to compare all data that can be reasonably compared (by having similar smoothing characteristics) and not only data that have a particularly high quality (small errors).

Section 4: Boundary layer

a)The theoretical error budget for the combined product is not in agreement with the experimental comparison (higher statistical errors and lower systematic ones). Also, in the abstract the authors mention that the combined products offer theoretically and experimentally better results. This is true for the sensitivity, but not for the error estimation shown, which could cause some confusion to the readers. The authors provide some causes to account for these discrepancies, but they have considered re-doing the error estimations with more realistic values. For example, is it enough to assume 5% and 0.1 rad for the ILS errors when the UNAM ILS is assumed as ideal?

Already in the abstract we will better specify what can be improved by using two measurements/ retrieval products instead of one measurement/retrieval product.

We think that the assumption of a 5% uncertainty in the ILS is realistic, also for the medium resolution instrument (recall that OPDmax is only 9 cm for the medium resolution instrument). However, this is only true if the retrievals work with the measured ILS. So far we used a nominal ILS for the medium resolution retrievals. Figure 1 reveals a loss of the modulation efficiency for our medium resolution instrument of 9% at OPDmax, which has so far not been correctly accounted for in the retrieval process. For this reason we will redo the retrievals by using the actually measured ILS. This retrieval improvement will reduce the bias between the in-situ data and the combined product and bring the observed bias in better agreement with the estimated systematic error.

In Section 4.3.2 we discuss three possibilities for the bias with respect to the in-situ data: O3 pressure broadening parameter, O3 continuum absorption, and the surface in-situ data being a poor reference for the whole boundary layer. We agree with the referee that we should in addition consider ILS uncertainties of the UNAM instrument as possible reason for the bias.

We think that the next step must be to perform O3 radiosonde measurements in coincidence with FTIR measurements in order to understand whether the bias between the surface in-situ and the combined FTIR data is an error or if it is actually real and can be explained by the fact that the surface in-situ data do not correctly represent the boundary layer.

b) As the authors stated, the combined product is very promising. I am interested in other possible applications of this product. Could it be used the other way round, ie, to reduce the tropospheric signal in the Atlzomoni ULTS data? Or to improve the tropospheric sensitivity of space-based sensors such as IASI?

We developed the method in line with the Rodgers formalism and it can be used whenever the combination of two remote sensing data increases the knowledge of the atmospheric state. An example could be to combine thermal nadir satellite data (low sensitivity close to the ground) with ground-based solar absorption data (good sensitivity close to the ground) or to combine thermal nadir infrared and short-wave infrared satellite data. This is a good point and we can add a brief comment in the "Summary and Outlook" Section.

An improvement of the Altzomoni data by using additional UNAM data seems not too promising, because the Altzomoni FTIR instrument is located at 4km and not affected by atmospheric variations below 4km (where the UNAM measurement can provide additional information). This means that the additional information given by the UNAM data will not be too useful for the Altzomoni data.

**Technical Comments**

Page 5, line 12. Please include reference for the WACCM model version 6.
Ok!

Page 7, line 14. Replace "in temperature" by "temperature".
Ok!

Page 10, line 8.Replace "subtropical" by "sub-tropical".
Ok!

Page 12, line 5. Replace "in situ" by "in-situ".
Ok!

Page 16, line 11. Replace "Eq. 10" by "Eq.(10)".
Ok!

---

## Author Response (AR1)

Dear Editor,

We made the revisions of the manuscript in line with the comments of the referees and our replies to these comments.

Major changes are:

- Reprocessing of the UNAM data using measured ILS instead of nominal ILS. This modification significantly reduced the bias between the combined boundary layer product and the boundary layer in-situ data.
- We added an additional subsection for explaining the working principle of the combined product in an intuitive way.
- We added two appendices with five additional Figures, in order to address the referees' comments on ILS and smoothing characteristics.

Please find attached a "latexdiff"-compiled pdf file that highlights all the modification with respect to the published AMTD version.

Best regards.

[revised manuscript text omitted]